# Endocrine Petrified Ear: Associated Endocrine Conditions in Auricular Calcification/Ossification (A Sample-Focused Analysis)

**DOI:** 10.3390/diagnostics14121303

**Published:** 2024-06-19

**Authors:** Ana Valea, Claudiu Nistor, Mihai-Lucian Ciobica, Oana-Claudia Sima, Mara Carsote

**Affiliations:** 1Department of Endocrinology, “Iuliu Hatieganu” University of Medicine and Pharmacy, 400012 Cluj-Napoca, Romania; ana74us@yahoo.com; 2Clinical County Hospital, 400347 Cluj-Napoca, Romania; 3Department 4-Cardio-Thoracic Pathology, Thoracic Surgery II Discipline, “Carol Davila” University of Medicine and Pharmacy, 020021 Bucharest, Romania; 4“Dr. Carol Davila” Central Military Emergency University Hospital, 010242 Bucharest, Romania; lucian.ciobica@umfcd.ro; 5Department of Internal Medicine and Gastroenterology, “Carol Davila” University of Medicine and Pharmacy, 020021 Bucharest, Romania; 6PhD Doctoral School, “Carol Davila” University of Medicine and Pharmacy, 020021 Bucharest, Romania; oana-claudia.sima@drd.umfcd.ro; 7“C.I. Parhon” National Institute of Endocrinology, 011683 Bucharest, Romania; carsote_m@hotmail.com; 8Department of Endocrinology, “Carol Davila” University of Medicine and Pharmacy, 020021 Bucharest, Romania

**Keywords:** calcification of auricular cartilages, petrified ear, ossification, endocrine, thyroid, adrenal, hormone, biopsy, cortisol, surgery

## Abstract

Petrified ear (PE), an exceptional entity, stands for the calcification ± ossification of auricular cartilage (CAC/OAC); its pathogenic traits are still an open matter. Endocrine panel represents one of the most important; yet, no standard protocol of assessments is available. Our objective was to highlight most recent PE data and associated endocrine (versus non-endocrine) ailments in terms of presentation, imagery tools, hormonal assessments, biopsy, outcome, pathogenic features. This was a comprehensive review via PubMed search (January 2000–March 2024). A total of 75 PE subjects included: 46 case reports/series (N = 49) and two imagery-based retrospective studies (N = 26) with CAC/OAC prevalence of 7–23% (N = 251) amid routine head/temporal bone CT scans. Endocrine PE (EPE): N = 23, male/female ratio = 10.5; average age = 56.78, ranges: 22–79; non-EPE cohort: N = 26; male/female ratio = 1.88, mean age = 49.44; ranges: 18–75 (+a single pediatric case).The longest post-diagnosis follow-up was of 6–7 years. The diagnosis of PE and endocrine anomalies was synchronous or not (time gap of 10–20 years). A novel case in point (calcified EPE amid autoimmune poly-endocrine syndrome type 2 with a 10-year post-diagnosis documented follow-up) was introduced. We re-analyzed EPE and re-classified another five subjects as such. Hence, the final EPE cohort (N = 50) showed: adrenal insufficiency was the most frequent endocrine condition (36%) followed by hypopituitarism (22%) and hypothyroidism (18%); 39% of the patients with adrenal failure had Addison’s disease; primary type represented 72% of all cases with hypothyroidism; an endocrine autoimmune (any type) component was diagnosed in 18%. We propose the term of “endocrine petrified ear” and a workflow algorithm to assess the potential hormonal/metabolic background in PE.

## 1. Introduction

Petrified ear (alternatively, named stony hard ear, stubborn petrified pinna or rigid pinna), an exceptional entity, stands for the calcification and/or ossification of the auricular cartilage [1,2]. The first report was done in 1886 based on a necropsy study that was performed by the anatomist Vincent Bochdalek [3] in a 65-year-old male [2,3,4], while some historical evidence from the Middle Age has also suggested this ailment [5].

Besides the idiopathic forms, multiple causes have been reported such as frostbite (hypothermia or cold exposure), local (mechanical) trauma (including pressure and friction), and inflammation (either localized or generalized). Various systemic diseases have been linked to this auricular condition, for example, sarcoidosis, chondromalacia, scleroderma, CREST syndrome (calcinosis, Raynaud’s phenomenon, esophageal dysmotility, sclerodactyly, and telangiectasia), relapsing polychondritis/perichondritis, arterial hypertension, alkaptonuria, polyarteritis nodosa, dermatomyositis, chronic tophaceous gout, and familial syndrome of cold hypersensitivity, as well as some neoplasia like facial naevi or chondroid syringoma [2,6,7,8,9,10,11].

Ossification, generally associated with a more severe presentation than seen in calcification, has been more frequently identified in congenital syndromes, for example, Albright’s hereditary osteodystrophy, osseous heteroplasia, congenital plate-like osteomatosis cutis and fibrodysplasia or myositis ossificans progressive etc. [4,7,12,13,14]. Exceptionally, iatrogenic forms (related to the radiation therapy for primary and secondary bone cancers), infectious aetiology (e.g., syphilitic perichondritis), and insect bite were incriminated in some instances of petrified ear [2,7,15,16,17]. 

Moreover, numerous endocrine illness were identified to be associated with this pinna anomaly such as autoimmune diseases (thyroiditis, adrenalitis causing Addison’s disease), hormonal deficiencies as seen in hypothyroidism or primary/secondary adrenal insufficiency (hypopituitarism may also include other hormones deficiency, and not necessarily an autoimmune background), hormones excess (acromegaly, thyrotoxicosis), and diabetes mellitus, as well as a wide area of calcium and/or parathormone (PTH) anomalies involving primary and secondary (renal) hyperparathyroidism, hypoparathyroidism, hypervitaminosis D, and pseudohypoparathyroidism. Among these endocrine issues, the adrenal failure of any type (adrenal- or pituitary-related) represents the most common form of endocrinopathy to cause the mentioned auricular lesions, and Addison’s disease is considered by the most authors the more frequent kind of adrenal failure (between the primary and secondary categories) [1,2,18].

The most common cause of petrified ear in the general population is frostbite. Some authors consider a certain age-related pattern (subjects older than 60 years seem at higher risk). Males are more prone than females to present a calcified pinna. The ear involvement may be unilateral or bilateral (this being more frequent). While auricular calcification/ossification remains exceptional, the human body globally may present calcium deposits in any tissue, underlying various local and general endocrine and non-endocrine mechanisms [2,3,4,5,6,7,8,9,10,11,12,13,14,15,16,17,18,19].

Currently, there is no specific protocol to assess the petrified ear, but a complex multidisciplinary panel of investigations is required. Until now, no distinct indications are listed to surgically approach the condition and also for general treatment, and thus the topic remains at individual decision in relationship with the ear changes and the underlying general condition (if any).

Our purpose was to highlight the most recent published data with concern to the petrified ear and associated endocrine (versus non-endocrine) conditions. We followed several aspects in terms of the clinical presentation, imagery tools, hormonal assessments, biopsy, outcome, potential pathogenic features, and proposed work-up algorithm to address the evaluation of a patient that is diagnosed with a rigid pinna amid an endocrine perspective. 

This was a comprehensive (narrative) review based on a PubMed search from January 2000 to March 2024 by using different combinations of terms such as “calcified ear” (alternatively, “petrified ear”, “calcified pinnae”, “stony ear”, and “calcification of auricular cartilages”). The (endocrine and non-endocrine) sample-focused analysis was based on freely available, full-length articles varying from case reports to original studies. We excluded reviews, non-human data, and non-English papers. A total of 49 subjects were identified with the diagnosis of petrified pinna amid 46 single case reports, one case series of two individuals, respectively of three subjects in addition to two studies (N = 26 patients with auricular calcifications/ossifications), hence, a total of 75 individuals diagnosed with petrified ear to whom a heterogeneous spectrum of parameters has been provided. The case reports and series were analyzed based on the original pathogenic perspective as being either endocrine-related (N = 23 patients) or no hormonal issue was considered to be involved in the presence of the petrified pinna by the original authors (N = 26). Discussion section introduces several connected issues in terms of potential pathogenic traits, a re-classification of some cases there were initially considered as being non-endocrine, and a proposed workflow algorithm to assess the hormonal component in patients with this auricular lesion, as well as a brief description of a novel clinical vignette on the particular matter of the endocrine petrified ear that is meant to add to the limited pre-existing data so far. The patient’ variable were retrospectively collected within a time frame of almost a decade. The Local Ethical Committee approved the retrospective data collection (as pointed out at the end of the article) (Figure 1).

## 2. Petrified Ears

### 2.1. Sample-Focused Analysis

#### 2.1.1. Endocrine Conditions

We identified a total of 23 papers enrolling 23 subjects diagnosed with any type of endocrine condition and auricular calcification/ossifications (of note, we used the terminology according to the original references) [1,6,18,20,21,22,23,24,25,26,27,28,29,30,31,32,33,34,35,36,37,38,39]. (Table 1).

According to our methods, the (endocrine) sample-based analysis (N = 23 single case reports) included papers across a 24-year timeframe of search that were published as follows: one paper (reporting a single patient) per year (N = 10 patients, reported in 2023, 2017, 2015, 2014, 2011, 2008, 2007, 2006, 2004, and 2002) and two articles (one subject per article) per year of publication (in 2022, 2021, 2019, 2016, and 2012, hence, a total of 10 individuals), while in 2009 three more single case reports were added to the general pool of data with respect to petrified ears (N = 3 patients). This sample-focused analysis showed a male to female ratio of 10.5 (N = 23, an average age of 56.78, ranges: 22–79 years); males (N = 21) had a mean age of 57 years (ranges from 22 to 79 years), and females (N = 2) had a mean age of 54.5 (aged of 52, respectively, of and 57 years) [1,6,18,20,21,22,23,24,25,26,27,28,29,30,31,32,33,34,35,36,37,38,39]. 

#### 2.1.2. Non-Endocrine Sample-Based Analysis

Additionally, we analyzed the published data with concern to the petrified ear that was confirmed in subjects without an actual diagnosis of a traditional endocrine ailment amid their medical records/history or during current admission (according to the original references). By applying the same mentioned methods, we further identified 22 single case reports (one subject with petrified ear confirmation per paper) [2,4,7,9,12,40,41,42,43,44,45,46,47,48,49,50,51,52,53,54,55,56,57], and one patient was reported via a case series of two that also introduced another family member with a concomitant endocrine condition [36], as well as a case series of three patients [44] (hence, a total of 23 distinct articles other than those cited at endocrine-based analysis; N = a total of 26 subjects diagnosed with petrified pinna). The timeline analysis showed one case per publication per year (in 2023, 2020, 2015, 2014, 2013, 2012, 2008, and 2005), two papers per year (one subject per paper) published in 2017 and 2009, respectively, and three papers per year (in 2019, 2016, 2011, and 2007; of note, the paper published in 2016 was a three-patient series [44]). 

This cohort (N = 26) had a male to female ratio of 1.88, mean age of adults (N = 25) was of 49.44 (ranges: 18 to 75) years; average age for the male sub-group (N = 17) was of 52.58 (range: 20 to 75) years, respectively, of female adults sub-group (N = 8) was of 42.75 (range: 18 to 73) years (and a single pediatric case of 9-year-old girl was identified [56]). Notably, some of these articles specified a normal hormonal profile, while others only classified the auricular involvement as being idiopathic or syndromic (or related to a systemic disease) without providing a specific hormonal work-up. We included in this second (non-endocrine) cohort the subjects that were not considered by the original authors as having a connection with the endocrine field, namely: idiopathic (N = 15), hypothermia- or trauma- induced (two individuals and, one subject, respectively), Primrose syndrome (N = 2), relapsing polychondritis (N = 3), osteoprotegerin-deficiency juvenile Paget disease type 1 (N = 1), and Keutel syndrome (N = 2) [2,4,7,9,12,36,40,41,42,43,44,45,46,47,48,49,50,51,52,53,54,55,56,57] (Table 2).

### 2.2. Clinical Detection 

The presentation seemed to be asymptomatic or mildly symptomatic and the detection was accidentally done during physical exam or imagery assessments. Unless a distinct phenotype was already described, the rigid pinna *per se* offered a low specificity with concern to the potential associated endocrine/non-endocrine conditions. Even an ossification associated with a calcification or displacing an auricular calcification may remain clinically undetected [15]. 

Physical examination revealed rigid (or stiff) ear cartilages; the rigid helices was also found by self-palpation [2,18]. Additionally, a subject may describe a progressive ear stiffness over the years or a difficulty to manually fold the ear lobe [2], a mild persistent or intermittent pain/otalgia [1] or a local pressure sensation/discomfort during sleeping position on the affected ear or during washing/cleaning [2,41]. The ear discomfort during sleeping might cause insomnia [12]. The skin was usually normal; other rare local cutaneous issues included: an erythema [4], a skin ulceration [12]; a mild helix hyperpigmentation (regardless the co-presence of an uncontrolled Addison’s disease with cutaneous and mucosal diffuse/extensive hyperpigmentation) [6]. 

Personal and family history might rarely provide clues for the pinna calcifications (such as autoimmune/non-autoimmune endocrine conditions) [2,36] or other non-endocrine (syndromic) diseases that may be pathogenically connected with this kind of ear involvement [44]. 

Typically, there was no hearing loss, but some patients may present probably an age-related hypoacusia (to some degree). The otoscope assessment of the external auditory canal and tympanic membrane was usually normal [2,18], but some data reported the fact that otoscope passing was not feasible due to calcifications/ossifications [7]. This might induce a conductive hearing loss (mostly, in auricular ossification rather than calcification) in addition to potential cerumen retention [18]. A single case of a pathological auricular fracture due to a petrified ear was reported in 2019 in a 52-year-old male [41]. 

Overall, according to the sample-focused analysis, unilateral ear involvement was found in 2/23 (8.69%) versus 6/26 (23.07%) subjects of the endocrine [1,6,18,20,21,22,23,24,25,26,27,28,29,30,31,32,33,34,35,36,37,38,39] versus non-endocrine [2,4,7,9,12,36,40,41,42,43,44,45,46,47,48,49,50,51,52,53,54,55,56,57] cohorts (of note, the unilateral or bilateral calcification/ossification of the pinna was assessed based on the clinical exam and/or imagery evaluation; hence, a total of 8 patients with unilateral ear lesions were confirmed, representing 16.32%). In the endocrine cohort, the ear complains (N = 8) on admission were local pain [1,18], detection/self-detection of rigid pinna [18,21,26,30,36,37,38], and hearing loss [18], hence the other 15 cases out of the 23 may be considered truly asymptomatic from the auricular perspective, representing 65.21% of the entire endocrine cohort. Notably, the asymptomatic pinna presentation did not exclude an uncontrolled underlying endocrine disease such as an acute adrenal failure. 

### 2.3. Imagery Assessments Amidst Identifying a Petrified Ear

To date, the first X-ray confirmation of an auricular calcification is from 1899, and it was performed by Wassmund [58]. Imagery evaluation might be the first step of identifying an auricular calcification when the procedure was performed for other (apparently unrelated) conditions, hence, respecting a true scenario of an incidental finding. On the other hand, the patient’ complains may be followed by the imagery procedures in order to confirm the pinna lesions [23]. 

The investigations helping the diagnosis were mostly X-ray and computed tomography (CT) scan. Whole body bone scintigraphy with 99m-Tc (Technetium) associated with single-photon emission CT (SPECT) was used in some instances. Synchronous Tc tracer uptake in other areas of the human body might reflect a general endocrine or syndromic condition (for instance, at the larynx and trachea, ribs, and intervertebral discs with ectopic calcifications) [1]. 

Temporal CT scan provided useful insights in terms of showing uniform hyper-densities (suggestive for auricular calcification) or radiolucent spaces within hyper-dense areas (suggestive for trabecular bone features amidst auricular ossification) [41]. A density of at least 1100 Hounsfield units at CT scan was prone for ossification type [1]. Alternatively, a 3D (dimensional) CT scan was used to confirm the petrified ear [22,25].

Additional pituitary exploration for hypopituitarism involves CT and/or magnetic resonance imagery (MRI) scans [23,27,30,32]. Three cases out of 23 with endocrine traits (representing 13.04%) had an empty sella [23,27,30], and one MRI scan identified a small pituitary gland in a 57-year-old lady with lymphocytic hypophysitis causing post-partum hypopituitarism with late onset [32].

In addition to the 49 patients amid our endocrine (N = 23) and non-endocrine (N = 26) sample-based analyses, we identified two retrospective studies [7,15] (of note, one of these articles also included a novel case report, as we already mentioned [7]). Both studies addressed the matter of calcified/ossified pinna from an imagery perspective since the condition was identified amid consecutive series of subjects who underwent different types of CT scans. The largest study on imagery features (but no specific data on the endocrine profile was included) was published in 2014 and it showed that, among 200 subjects with head CT scans, 19.5% (N = 39) of them had calcifications of the cartilage from the external ear (cartilage of the auricle and external auditory channel), mostly being incidental and asymptomatic. Gossner et al. [15] showed that CT scan remains the best option strategy for the diagnosis of the petrified pinna [15]. In addition to these 14/39 patients who had auricular cartilage calcifications/ossifications [15], Aw et al. [7] introduced 51 consecutive patients who underwent temporal bone CT scans and 12/51 of them had foci or extensive auricular calcifications [7]. 

This mentioned subgroup of individuals that have been identified across these two studies (N = 26) involved a rather unexpected high rate of auricular involvement of calcification kind (of 23.53%, N = 12/51) across temporal bone CT assessments [7], respectively, across head CT scans (of 7%, N = 14/200) [15] thus suggesting that pinna calcification/ossification might be underdiagnosed. (Table 3)

Overall, the data coming from our sample-based analysis via isolated case reports (N = 49 subjects) showed that X-ray and/or CT confirmation was done in all cases (no differences were noted in imagery approaching the patients with an endocrine [1,6,18,20,21,22,23,24,25,26,27,28,29,30,31,32,33,34,35,36,37,38,39] versus non-endocrine [2,4,7,9,12,36,40,41,42,43,44,45,46,47,48,49,50,51,52,53,54,55,56,57] background). As limitations of the imagery evaluation we mention several factors such as the speed of diagnosis, associated cost of performing new assessments, resource availability, and technical complexity of the imaging techniques depending on the health care center. 

### 2.4. Hormonal Panel in Rigid/Petrified Pinna 

The hormonal assessments were provided in the endocrine cases [1,6,18,20,21,22,23,24,25,26,27,28,29,30,31,32,33,34,35,36,37,38,39], but, also, as mentioned, in non-endocrine cases (to some extent) [2,4,7,9,12,36,40,41,42,43,44,45,46,47,48,49,50,51,52,53,54,55,56,57]. Despite large variations in reporting an endocrine profile, the main evaluation included:A.Exploring the thyroid function in terms of TSH (thyroid-stimulating hormone) and FT4 (free thyroxine) in association with the thyroid antibodies (anti-thyroperoxidase and anti-thyroglobulin) that are suggestive for an autoimmune thyroid disease, particularly, Hashimoto’s chronic lymphocytic thyroiditis was mandatory. Notably, both primary and secondary hypothyroidism was reported amidst the diagnosis of petrified pinna [18,23].B.The pituitary function assessment stands for the hypopituitarism testing and/or hormonal excess (particularly, acromegaly) [1,58].C.The evaluation of ACTH (adrenocorticotropic hormone)-cortisol axis (baseline morning ACTH and plasma cortisol) also included dynamic testing to address the confirmation of a primary or secondary adrenal failure. Even mild forms of deficiency should be carefully taken into consideration [59]. Corticotropin-Releasing Hormone (CRH) test to assess ACTH deficiency (intravenous 100 µg CRH followed by ACTH and cortisol assays after one hour) was used to confirm an isolated ACTH deficiency across an ACTH value of 2.7 pg/mL and cortisol of 0.03 µg/dL during test [25]. Cosyntropin stimulation testing confirmed primary adrenal insufficiency in the case of a 47-year-old male reported by Sedhai et al. [23] (a value of plasma cortisol of 4.7 µg/dL after 60 min since injection). Negative results of the test with undetectable plasma cortisol suggested a secondary adrenal insufficiency in a 45-year-old male with bilateral foci of auricular ossifications (and negative endocrine autoimmunity) [28]. ACTH stimulation test remains a practical approach of the adrenal insufficiency diagnosis (including partial or latent forms) amid the presence of the rigid ears according to the general endocrine practice [59,60,61,62,63].D.Another important issue included the evaluation of the metabolic features such as diabetes mellitus (four subjects within the endocrine cohort [1,21,34,38], but, also, two other patients in the non-endocrine cohort [46,50], had the condition, as well as one case of hypercholesterolemia [41]).E.Also, the renal function and calcium–-phosphorus metabolism assays, including 25-hydroxyvitamin D and PTH (parathormone) were useful in relationship with potential petrified pinna-associated causes. For instance, Weiss et al. [12] showed a bilateral (started as unilateral) case of auricular calcification in a 69-year-old male who was known with chronic kidney disease due to high blood pressure. In addition to the renal dysfunction assays, vitamin D deficiency (a level of 25-hydroxyvitamin D of 16 ng/mL) was confirmed [12].

The admission (which allowed the confirmation of the rigid/petrified pinna in patients with endocrine diseases) was done for various purposes as following: ▶self-palpation of rigid pinna and bilateral stenotic external ear canals (N = 1) [21]▶progressive rigid pinna and hearing loss (N = 2) [6,18]▶intermittent ear pain (N = 1) [1,34]▶progressive rigid pinna (N = 2) [36,37]▶progressive rigid pinna and recurrent weakness due to adrenal insufficiency (N = 1) [22]▶acute pericarditis (N = 1) [23]▶acute adrenal insufficiency (N = 5) [25,28,29,31,32]▶ulcerative colitis (N = 1) [20]

Overall, we identified several types of endocrine- and auricular-related diagnosis patterns with respect to the timeline of identification: ▶Generally, the complains of the rigid pinna (intermittent pain or pain during certain sleeping positions) or the self-palpation of the rigid helix were presented for years before the actual diagnosis of the petrified ear [1,18,26,34]; the maximum period of time was of 20 years [18] in the endocrine cohort, and these patients had a certain hormonal anomaly that may be associated or not with the auricular lesion. ▶A concomitant diagnosis of the endocrine disease (and other co-morbidities) and of the calcified pinna was established [20,23,30,32].▶Prior history of an endocrine condition was followed by the self-detection of a rigid pinna [22,26]; the longest time frame between the endocrine diagnosis and the petrified ear diagnosis was of 20 years [22].▶Prior endocrine disease with clinical detection of the petrified ear amid hospitalization for these already known hormonal issues [6,27,31,33,35]▶Detection of the rigid pinna was followed by an endocrine diagnosis [28,36,37,38]; the maximum length of this time window was of 6 years [37], respectively, of 10 years [38] (Figure 2).

Notably, the hormonal panel was essential, even as a retrospective diagnosis; for instance, Machado et al. [32] published the first case of postpartum hypopituitarism due to autoimmune hypophysitis, but 15 years after the patient delivered her baby. She was admitted as an emergency underlying an acute adrenal failure associated with central hypothyroidism and intact gonadal axes (that helped the differential diagnosis from a Sheehan’s syndrome) [32].

Also, a severe presentation was registered in one case due to acute complications of the autoimmune endocrine conditions. Sedhai et al. [23] reported a male of 47 year-old who was admitted as an emergency for an episode of acute pericarditis; he presented polyserositis due to prior undiagnosed adrenal insufficiency and hypothyroidism. Addison’s disease panel included a plasma morning cortisol of 2 µg/dL that was highly suggestive; the plasma cortisol following Cosyntropin stimulation was of 4.7 µg/dL (after one hour since Cosyntropin administration); plasma morning ACTH was elevated (of 220 pg/dL). Primary autoimmune hypothyroidism confirmation was done via an increased TSH (of 16.43 mUI/L) and low T4 (0.93 mUI/L) with positive serum antibodies against thyroid, namely anti-thyroperoxidase and anti-thyroglobulin antibodies. Of note, the hormonal dysfunctions were complicated with a low sodium and high potassium. The patient also presented inflammatory syndrome in terms of high C reactive protein (of 196 mg/dL) and erythrocyte sedimentation rate (of 38 mm/h). Other rheumatologic assays such as rheumatoid factor, anti-SSA (anti-Sjögren’s-syndrome-related antigen A) autoantibodies, and anti-SSB antibodies (anti-Sjögren’s-syndrome type B autoantibodies), etc. were found negative. Hormonal replacement massively improved the clinical (endocrine) picture, but not the auricular elements [23]. Generally, serositis may accompany autoimmune endocrine conditions and symptomatic pericardial effusion has been reported mostly due to severe deficiency of the thyroid hormones potentially associated with reduced levels of cortisol that also causes hypotension and shock; nevertheless the differential diagnosis with other causes of pericarditis/pleural involvement in subjects with an endocrine background should be performed [64,65,66]. Recently, COVID-19 pandemic turned out to be a trigger of various autoimmune conditions the infection itself or the vaccine against coronavirus might aggravate an acute adrenal insufficiency in prior undiagnosed subjects or even in previously treated patients [67,68,69]. 

Remarkably, two cases of acromegaly (N = 2/23) were reported in 2022 in a 43-year-old male [1] a 58-year-old man [58]. Acromegaly caused by a pituitary GH (growth hormone)-producing macroadenoma was confirmed in a diabetic adult who underwent hypophysectomy (two times) and gamma knife therapy and then started somatostatin analogues. He was offered (in addition to metformin for diabetes mellitus) replacement therapy with daily levothyroxine and hydrocortisone for hypopituitarism. He had intermittent bilateral auricular pain for a few years (before the actual recognition of acromegaly, diabetes, and associated hypopituitarism). CT scan showed bone-like density in both external ears (initially, X-ray scan pointed out the same aspect). At the moment of the petrified ear diagnosis, diabetes and acromegaly were only partially controlled and plasma morning cortisol was low, at 3.1 µg/dL (suggestive for an adrenal insufficiency) [1]. 

The panel of non-endocrine diseases in poly-endocrine autoimmune syndromes should be taken into consideration with respect to petrified ear-related background. Zhao et al. [20] reported a combination of endocrine and non-endocrine autoimmune traits (autoimmune thyroid disease and ulcerative colitis) [20]. Further awareness of the risk of having new clinical entities across lifespan that are caused by common autoimmune pathogenic loops is mandatory [20,70,71,72,73]. On the other hand, the thyroid involvement is part of the extra-intestinal endocrine disease in ulcerative colitis [74]. To conclude, testing the panel of the autoimmune rheumatologic and gastroenterological conditions seems equally important in order to identify a non-endocrine condition such as polychondritis, autoimmune colitis, and systemic sclerosis that may be isolated or be part of an autoimmune poly-endocrine syndrome [12,20]. Currently, the specific strategy of assessment protocols had great variations across the published data in the matter of auricular calcifications. 

### 2.5. Histological Traits

The confirmation of the auricular calcification/ossification was based on the histological analysis via incisional/excisional biopsy or local surgery. Both cortical and trabecular bone elements were found in the ossified type of petrified pinna. Dermis and epidemis was normal as accessed by biopsy [1,4]. Pathological report is not mandatory for the diagnosis since radiologic/imagery findings are highly suggestive (mostly based on the CT scans). Of note, the histological testing might provide the distinction between calcification (fibrocartilaginous tissue) and ossification (lamellar bone and bone cells such as osteocytes), which that is also feasible through CT [2,4,18]. The calcification underlines deposits of amorphous, insoluble calcium salts within the local tissue. Local inflammation (that stands as a contributor/trigger to calcium deposits) may be found at pathological report, typically in dystrophic rather than metastatic cases. Calcification in petrified ears should be distinguished from isolated nodules found in calcinosis cutis, which is more common in pediatric populations [75,76,77]. Some authors suggested that calcification may turn into ossification amid the local action of different triggers such as trauma or frostbite [4]; thus poly-factorial etiology may be present. Notably, Bochdalek initially showed an ectopic auricular ossification [3]. 

Overall, the endocrine sample-based analysis [1,6,18,20,21,22,23,24,25,26,27,28,29,30,31,32,33,34,35,36,37,38,39] pointed out 12 cases of (partial or complete) ossification [1,6,18,20,22,26,28,32,33,34,38,39] out of the 23 (either confirmed by histological report, radiologically or both) thus the rate of 52.17% was higher than estimated by prior published data [4]. The non-endocrine cohort [2,4,7,9,12,36,40,41,42,43,44,45,46,47,48,49,50,51,52,53,54,55,56,57] revealed 16 subjects with auricular ossificans (foci or extended lesions) which represented 61.53% of the cohort, also, higher than it has previously been reported [4,12,41,42,44,46,47,50,51,52,53,54,55,57]. 

### 2.6. Practical Approach and Expected Outcome

This section includes three main aspects. Firstly, as mentioned, there is the question of routinely performing a biopsy which currently is selectively indicated according to an individual decision by a multidisciplinary team. We identified four cases (N = 4/23, representing 17.39% of the endocrine cohort [28,32,34,38], two cases with trabecular pattern [28,38], and one with Haversian features [32]), respectively, five subjects (5/26, representing 19.23% of non-endocrine cohort [4,36,42,50,53]) with incisional (mostly) or excisional biopsy. Of note, in another case, biopsy was attempted, but was not successful due to severe ear rigidity [55]. 

Secondarily, in people with petrified pinna and endocrine conditions, we might expect that, in case of a clear pathogenic connection, the control of the hormonal anomaly limits the auricular lesion progression, an aspect which was not confirmed so far. As mentioned, this type of ear damage may be identified as first sign of an endocrine anomaly or during follow-up in longstanding (less or more controlled) hormone-associated conditions; as mentioned, sometimes the rigid helix is presented years before the endocrine illness is actually recognized [18]. Of note, Addison’s disease may be well controlled under adequate hydrocortisone and fludrocortisone therapy at the time when the hardening of the helix progresses [26]. Early identification of a petrified ear might pinpoint the need of an endocrine check-up or a general medical assessment with concern to other non-endocrine (potentially)—related issues. The most important endocrine condition, adrenal insufficiency, might complicate with an acute form that represents a life-threatening situation thus, adequate recognition starting even from unusual clues such as calcifications of the auricular cartilages improves the overall outcome [1,2]. To conclude, there is no pathogenic treatment a therapy to reverse calcification so far. However, the adequate treatment of the underlying endocrine or non-endocrine disease might help the pinna lesion on theoretical ground. The longest duration of follow-up after the confirmation of the petrified pinna was of 6–7 years according to our research [21,25]. For instance, the 22-year-old male who was detected by Taguchi et al. [25] with secondary adrenal insufficiency and petrified ears was followed for 6 years with a good clinical outcome, but no remission of the auricular lesion upon the endocrine disease control through oral hydrocortisone was identified [25]. 

Thirdly, there is the issue, yet an open issue, of performing surgery for petrified ears. Unless the disease is symptomatic and it severely impairs the overall quality of life or a local malignant lesion is co-detected, local surgery is not indicated (for example, procedures such as wedge resection of the calcified lesion at the level of external cartilage or concha reduction technique) [2]. Generally, skin graft, second intention healing, and trans-cartilaginous flaps are less feasible due to the anatomical particularities of the pinna in this instance; they are used for other body sites or for the reconstruction of the pinna in microtia [18,78,79,80,81,82]. Currently, auricular cartilage tissue engineering is part of the modern regenerative medical approach for an adequate reconstruction of the pinna amid different underlying conditions [83,84,85], but, petrified ear does not seem to be one of them. 

The cited papers according to our methods did not specify any surgical intervention in most cases in addition to the general recommendations and life style intervention such as avoiding the local trauma and frostbite as well as the use of mild non-steroidal anti-inflammatory drugs and an orthotic pillow to prevent a local trauma during sleeping [12]. A part from the mentioned case of excisional biopsy in a 75-year-old male with Addison’s disease and diabetes mellitus [34], no other surgical approach was mentioned in the endocrine cohort [1,6,18,20,21,22,23,24,25,26,27,28,29,30,31,32,33,34,35,36,37,38,39], while in the non-endocrine cases [2,4,7,9,12,36,40,41,42,43,44,45,46,47,48,49,50,51,52,53,54,55,56,57], one individual refused surgery [41], meatoplasty was performed in a 49-year-old male with idiopathic ossification of the cartilaginous auricle and external auditory canal [54], and in a 72-year-old male, after biopsy did not succeed, a wedge excision of the rigid upper rim of the right auricle was done with a clinical improvement of the unilateral petrified ear that involved a 7-month history of swollen, rigid pinna that was painful during sleep [55].

## 3. Discussion

### 3.1. Sample-Based Data: Current Evidence versus Historical Evidence 

We identified a total of 75 individuals across the most recent isolated case presentations and two retrospective studies, and 23/75 of these patients had an endocrine background [1,6,18,20,21,22,23,24,25,26,27,28,29,30,31,32,33,34,35,36,37,38,39] that may be placed in relationship with the development of pinna calcifications and/or ossificans according to current understanding, while the other 26/75 subjects were initially seen as idiopathic or (non-endocrine) syndromic features [2,4,7,9,12,36,40,41,42,43,44,45,46,47,48,49,50,51,52,53,54,55,56,57]. Another 26/75 patients were diagnosed with this ear lesion amid radiological screening (but no particular hormonal work-up was provided), with this sub-group being collateral to the actual sample-focused analysis that mainly targeted the endocrine versus non-endocrine traits [7,15]. 

These 75 subjects suffering from petrified ear were identified across a 24-year (23 years and three months) search according to our mentioned methods. This stands for one of the largest and most detailed analyses on published modern data with respect to the prior publications. For instance, Recalcati et al. [18] specified in a paper from 2021 that probably only 150 cases have been reported so far (the same number is also mentioned by Shah et al. in 2019 [4]), while Chan et al. [50] raised up the idea of 140 cases of petrified ear and only 18 of them were auricular ossifications (in 2011) [50]. Similarly, Alsey et al. [49] showed in 2012 that fewer than 160 cases had been reported since the first identification in 1866 [49]. 

Chan et al. [50] remarked that most cases were idiopathic with (potential) identifiable contributors such as hypothermia exposure or endocrine co-morbidities; the male to female ratio was of 18 to 5, and the mean age at diagnosis was of 57 years, while 70% of them were bilateral lesions [50]. Our endocrine sample-focused analysis showed a male to female ratio of 10.5 (N = 23) with an average age of 56.78 years [1,6,18,20,21,22,23,24,25,26,27,28,29,30,31,32,33,34,35,36,37,38,39] and, respectively, a male to female ratio of 1.88, with a mean age of adults (N = 25) was of 49.44 years in addition to one pediatric case [2,4,7,9,12,36,40,41,42,43,44,45,46,47,48,49,50,51,52,53,54,55,56,57]. Moreover, we found a rate of unilateral ear involvement of 16.32% (higher for non-endocrine versus endocrine cases 23.07% versus 8.69%). 

The ossification elements were more frequently confirmed than in previously reported data, of 52.17% and, respectively, of 61.53%, respectively. A potential explanation may be related to the concept expansion in the field of petrified pinna and a higher awareness amid different practitioners of the radiological and histological features. Similarly, Weiss et al. [15] provided in 2017 a review of the English-published papers and pointed out a pathological confirmation in 20 cases with auricular ossification (not calcification), between 1890 and 2011 (a rate of 1/20 patients included one case of Addison’s disease associated with diabetes mellitus) [15]. Calderón-Komáromy et al. [28] mentioned in 2015 about 160 subjects diagnosed with petrified ears of any cause via prior literature search; they specified about 20/160 cases of auricular ossifications and only 3/20 patients were diagnosed with adrenal insufficiency (one individual had a secondary type of adrenal failure due to postpartum hypophysitis and the other two patients had Addison’s disease) [28]. 

Historically, the first articles across our PubMed search were published in 1955, respectively, 1960 (regarding the connection to Addison’s disease) [86,87], in 1961 (with concern to sarcoidosis-related hypercalcemia) [88], in 1963 (hypopituitarism as endocrine background in petrified ear) [89], and in 1943 (auricular ossification in acromegaly) [90]. The earliest (large) reviews mentioned 119 patients (and 14 out of these 119 subjects had Addison’s disease) [91], respectively, 65 individuals with auricular calcification/ossifications [92]. This timeline perspective showed us that early identification of the condition was not necessarily followed by an increased number of published cases, but rather by a more complex panel of investigations to highlight potential pathogenic connections. However, the tide correlation between the evolution of the petrified ear and the underlying systemic condition outcome could not be established, neither a specific therapy in this particular matter was not developed. We appreciate that, as mentioned by the two retrospective studies [7,15], the petrified pinna diagnosis might be actually underestimated, thus a multidisciplinary awareness might help.

### 3.2. Potential Pathogenic Features in Petrified Ear

Auricular ossification stands for a more severe form when compared to calcification, whereas the physiological elastic cartilage is completely or partially replaced by the bony tissue. This may co-exist with a calcification. We found no association between the ossificans pattern and demographic or hormonal characteristics of the patients [1,4,6,12,18,20,22,26,28,32,33,34,38,39,41,42,44,46,47,50,51,52,53,54,55,57]. It might suggest that local elements including paracrine contributors should be taken into account. One the other hand, inflammation or even biochemical, metabolic, and hormonal anomalies might not be stationary over the years and the moment of admission (and associated assessments) might not capture the true essence of such transitory changes and their actual impact of the ear cartilage. For instance, hypercalcemia (as a potential contributor to metastatic calcifications) across a life span with Addison’s disease may be registered during an acute episode (acute adrenal insufficiency), severe dehydration or renal dysfunction. Yet, this is not a constant biochemistry result in such patients [93,94,95]. Thus, transitory hypercalcemia and even hyperphosphatemia in patients with mineral metabolism anomalies (including chronic kidney disease) and acromegaly may play an essential pathogenic role with respect to the auricular calcifications [32,96,97]. 

Generally, the spectrum of calcified/ossified pinna involvement varies from calcification to ossification, including foci of bony tissue among calcified cartilages. Some authors suggested that ossification is the end stage of calcification [6,15]. These tissue calcifications are generally classified as being either dystrophic (calcium deposits, but without general anomalies of the mineral metabolism, for instance, due to endothelial lesions and microangiopathy as seen in uncontrolled diabetes) and metastatic (accompanied by anomalies of the calcium–phosphorus metabolism that might cause calcium deposits virtually in any tissue) [1,98]. Other local mediations of ectopic calcifications are bone morphogenic proteins (BMP) and bone-derived growth factors [41,99,100,101]. BMP-5 expression increases during chondrocyte differentiation in vivo and in vitro, and it promotes proliferation and cartilage matrix synthesis in primary chondrocyte cultures [102,103,104]. On the contrary, CCN [cysteine-rich angiogenic protein 61 (CCN1/CYR61), connective tissue growth factor (CCN2/CTGF), and nephroblastoma overexpressed (CCN3/NOV)] family, particularly, CCN2/CTGF stands for an atypical growth factor for chondrocytes/cartilages that prevents local hypertrophy or calcification, and local defects of this system might be involved in abnormal calcium deposits [105]. 

As mentioned, most calcified ears may be found in these hormone-related diseases without any obvious anomaly of the serum/urinary calcium and/or phosphorus, and the underlying mechanisms remain fairly unknown. Other proposed pathogenic traits that do not involve calcium/PTH pathways and diabetic small vessels’ damage, relate to proliferative and degenerative cartilaginous elements due to excessive GH and/or IGF1 (Insulin-like Growth Factor 1) in acromegaly [106]. Of note, in acromegaly, secondary (endocrine) diabetes, hypopituitarism (particularly, secondary adrenal failure and central hypothyroidism) have been reported in relationship with the petrified ears, but the exact signal transduction loops are not fully understood [1,58].

Older hypotheses that are currently unconfirmed incriminated the therapy with 11-desoxycorticosterone acetate for Addison’s disease as a pathogenic element for auricular calcifications; nowadays, petrified pinna is found in patients who were prior undiagnosed and untreated for adrenal failure, thus, this iatrogenic component is no longer taken into account [87]. Despite adrenal insufficiency representing the most common endocrine entity associated with the petrified pinna, ACTH does not seem to be the incriminating factor since both situations (with low and increased ACTH) were described in petrified ear. In fact, Addison’s disease- and hypopituitarism-related adrenal failure might pose different mechanisms of pinna lesions, apart from common low cortisol effects and general homeostasis implications [25]. Moreover, there is functional evidence between the melanocortin and opioid receptors, on one hand, and the osteoarticular system, on the other hand, including cartilages. Whether an elevated ACTH might act on cartilages, bone and synovial tissue as similarly seen with other proopiomelanocortin (POMC), derivate peptides such as α-MSH (melanocyte-stimulating hormone) or β-endorphin are further to be studied [25,106,107,108,109,110,111]. Of note, α-MSH (amid negative feedback-based POMC release due to low cortisol in primary adrenal insufficiency) may modulate cell adhesion as well as the inflammatory process that involve synovial fibroblasts [107,109]. 

Additionally, whether hardening of the pinna has a familial pattern in endocrine diseases (as seen in autoimmune syndromic endocrine diseases) is still an open issue. Thomson et al. [36] reported two family members who displayed bilateral calcifications of the ear cartilages since early age. The father (a 70-year-old male) had a history of hypothermia exposure, but he was also diagnosed years before with autoimmune hypothyroidism and Addison’s disease that may be contributors to petrified pinna, too, as far as we know by now. On the other hand, his daughter, a 30-year-old female showed, as well, a gradual stiffness of the auricular cartilages since childhood, but screening testing showed no endocrine condition (including an adequate response to short Synachten test to exclude a partial or a prior unidentified primary adrenal failure) [36]. 

As specified, the non-endocrine cohort (N = 26) according to our methods highlighted several syndromes/systemic diseases and some of them might include hormonal anomalies; if they should be placed on the list of “endocrine petrified ear” is still a matter of debate. For instance, Arora et al. [40] recently revealed five patients diagnosed with Primrose syndrome (an autosomal dominant disease whereas the patients harbor pathogenic variants of the *ZBTB20* gene on the chromosome 3q13.31 [112,113] with a complex clinical presentation, including intellectual impairment, macrocephaly, hearing loss, and spine involvement); the authors considered that calcifications of the pinna should be regarded as part of the phenotype. Interestingly, the syndrome might include metabolic and endocrine traits such as glucose metabolism anomalies, and short stature [40]; congenital hypothyroidism was reported since 2016 in association with two novel missense pathogenic variants (Ser616Phe and Gly741Arg) [114]. 

Another finding is reflected by the consideration of pinna ossification in patients diagnosed with osteoprotegerin-deficient juvenile Paget’s disease type 1 (which is characterized by an elevated bone alkaline phosphatase in association with bone pain and local deformities due to an increased bone turnover; a disease underlying pathogenic variants of the *TNFRSF11B* gene that encodes osteoprotegerin [115,116,117,118]). Gottesman et al. [44] reported a four-patient case series diagnosed with the condition, and three of them experienced petrified ear even from an early age in two of these subjects [44]. 

Keutel syndrome, an autosomal recessive hereditary illness, includes diffuse cartilage calcifications in the larynx and tracheobronchial tree associated with petrified ear underlying calcification or ossification, brachytelephalangism, and peripheral pulmonary stenosis. This is caused by the anomalies of the matrix GLA extracellular protein (MGP) due to pathogenic variants of the *MGP* gene. MGP stands as a tissue calcification inhibitor. The first description was done by Keutel in 1971 and three decades later the pathogenic connection with *MGP* gene (loss of function) was identified [119,120,121,122]. According to our methods, two such cases [47,56] introduced a petrified pinna that seems related to the underlying matrix anomalies, not to a specific hormonal interplay. Specifically, auricular calcifications were described by Parmar et al. [56] in a 9-year-old girl (the youngest patient amid our literature search), while Khosroshahi et al. [47] reported a case series of four sisters coming from a consanguineous family and one of them (who was prior reported as a single case presentation [123]) died at the age of 37 after cesarean; she had peripheral pulmonary stenosis, typical facies, calcifications of the epiglottis and trachea, brachytelephalangism, and bilateral auricular ossifications [47,123]. 

A distinct specification (other than potential hormonal implications), but taking into account an autoimmune background, involves the presence of relapsing polycondritis in three subjects, one 59-year-old male [43], and two young adult females, of 20 [48], respectively, of 29 [45] years. This stands for a rare clinical entity that shows recurrent episodes of inflammation at the level of cartilages with direct anatomical and functional consequences [124,125,126,127,128,129]; under these circumstances, recurrent auricular chondritis with local infiltration of the inflammatory cells, and the cartilage degeneration might associate calcifications across life span [43,45,48]. 

### 3.3. Clinical Vignette: Endocrine Petrified Ear 

This is a novel case on point to add to the limited number of publications series with concern to the autoimmune poly-endocrine syndrome. To our aware this is the longest duration of documented petrified ear after first identification of an endocrine-based auricular calcification (of 10 years). A 52-year-old smoker male was admitted (in 2024) to address an endocrine check-up of previously diagnosed and treated autoimmune conditions, namely, primary adrenal failure (since the age of 14), and chronic Hashimoto’s thyroiditis with primary hypothyroidism (since the age of 42). A decade ago, he was admitted for an adrenal crisis (with normal kidney function and mineral metabolism), and a clinical exam showed bilateral rigid pinna (without hearing loss) underlying bilateral calcifications at CT scan (at that moment, a mild autoimmune hypothyroidism was firstly detected, and levothyroxine replacement therapy was offered to the patient). While no specific therapy was initiated for the ear lesion, he continued the hormone replacement therapy with a poor adherence to medication and many episodes of relapsing adrenal crisis. (Table 4). 

Currently, the clinical and imagery features of the petrified ear were stationary (Figure 3) .

Of note, the patient also associated persistent hypercholesterolemia, Gilbert syndrome, and, recently, hyperuricemia (no gout). He also had a MRI confirmation of a partial empty sella with no hypopituitarism (neither hypogonadism of other type was found; he had a normal IGF1, as well as 25-hydroxitamin D, PTH, and glucose profile). Lifelong surveillance is mandatory. The endocrine therapy was adjusted; no biopsy or surgery was recommended for the auricular lesions. 

A similar case was reported in 2021 by Recalcati et al. [18]; this was a 71-year-old male with local complains due to bilateral auricular stiffness; he had a history of almost two decades with concern to treated hypothyroidism and chronic adrenal insufficiency. This might be related to the autoimmune polyendocrine syndrome type 2; however, on the left side, the gentleman also suffered from a squamous cell carcinoma of scapha that required resection surgery followed by two-stage reconstruction [18]. 

### 3.4. Pinna Calcifications/Ossificans and Various Endocrine Conditions 

Endocrine diseases or hormonal imbalances may lead to calcified pinna, but their significance and management remain incompletely understood. This novel case as well as at least three reported patients [18,23,36] suffered from Schmidt’s syndrome. The large group of autoimmune poly-endocrinopathies stands as a positive diagnosis or a differential diagnosis in each new patient confirmed with auricular calcifications and/or ossifications. Autoimmune polyendocrine syndrome, firstly described by Schmidt in 1926 (hypothyroidism plus adrenal insufficiency) represents a combination of at least two autoimmune conditions. There are four types depending on the underlying diseases, genetic background, and inheritance pattern, as well as the age at onset. Type 1 (monogenic, autosomal recessive) typically manifests early, with affected individuals carrying pathogenic variants of the AIRE gene. The most important clinical elements are Addison’s disease, hypoparathyroidism (these being the most common endocrine features), and cutaneous/mucosal candidiasis with ectodermal dystrophy. Chronic autoimmune thyroiditis, hypogonadism, and type 1 diabetes mellitus are the second most frequent endocrine group of ailments. Type 2 (which is polygenic, and it is more frequently found than type 1, being associated with a complex hereditability profile) mostly affects adults, females being more prone than males; the onset is typically within the third or the fourth decade of life. The most common endocrine traits include autoimmune thyroid disease and Addison’s disease (Schmidt’s syndrome), but premature ovarian failure (autoimmune hypogonadism) and type 1 diabetes are frequently found, too. Type 3 stands firstly for not involving adrenal glands, but for having the combination of an autoimmune thyroid disease with another entity such as type 1 diabetes (type 3A), chronic atrophic gastritis, and/or pernicious anemia (type 3B) or vitiligo, alopecia, and myasthenia gravis (type 3C). Type 4 introduces other associations than the mentioned types [130,131,132]. 

Further on, we analyzed the endocrine sample-based cohort [1,6,18,20,21,22,23,24,25,26,27,28,29,30,31,32,33,34,35,36,37,38,39] in addition to this present case and another five cases from the initial non-endocrine cohort [2,4,7,9,12,36,40,41,42,43,44,45,46,47,48,49,50,51,52,53,54,55,56,57] which we re-classified due to the co-presence of endocrine-metabolic traits that potentially may be part of a pathogenic association (yet, this remains an open issue) as following: vitamin D deficiency and chronic kidney disease that potentially bring a large of calcium and phosphorus anomalies (even transitory or responsive to medical/surgical interventions for renal hyperparathyroidism or dialysis, but this still might be a potential risk factor for tissues calcifications) [12], hypercholesterolemia [41] (this may be interpreted as a risk factor for calcifications as seen in vascular/endothelial calcification [133]), diabetes, chronic renal disease, and gout [46], diabetes [50], and hypogonadism [52]. To conclude, a total of 50 cases (in addition to the novel case report) were included in the final analysis and 29 patients had on admission or in their medical history any type of endocrine disturbances, and we propose the term of “endocrine petrified ear” for this final endocrine cohort. (Table 5).

Endocrine petrified ear represented 58% (N = 29) of the fifty cases (including this novel vignette) which we have analyzed according to our methods. The co-presence of the hormonal/metabolic anomaly does not necessarily mean a pathogenic connection but, considering the current level of statistical evidence, awareness is necessary. Of note, one patient might present with more than one endocrine/metabolic issue. We identified the followings (the current case was included, too): six subjects with diabetes [1,21,34,38,46,50]; four cases with empty sella [23,27,30], respectively, with hypogonadism of hypo- [21,22,29] or hyper-gonadotropic type [52]; three cases with chronic kidney disease [12,37,46]; two patients with hypercholesterolemia [41], respectively, with acromegaly [1,21], vitamin D deficiency [12,30], and osteoporosis [50,52]; one patient with primary hyperparathyroidism [6], renal hyperparathyroidism [37]; and pseudohypoparathyroidism [35]. The most frequent kinds of endocrine issues were adrenal failure, hypothyroidism, and hypopituitarism. Adrenal insufficiency was diagnosed in 18 patients (representing 36% of all patients with endocrine petrified ear) [1,18,22,23,24,25,26,27,28,29,30,31,33,34,36], hypothyroidism was diagnosed in nine subjects (18% of the cases with endocrine petrified ear) [1,18,20,23,27,30,32,36], 11 people (22%) were confirmed with hypopituitarism underlying any type of hormone deficiency (mostly, of ACTH, and TSH, but, also, gonadotropes and GH) and cause (such lymphocytic hypophysitis, post-traumatic, and iatrogenic) [1,22,24,27,28,29,30,31,32,39] (Figure 4).

Among the cases of adrenal insufficiency (N = 18, 36%), only 39% were Addison’s disease and the others (N = 11, representing 61% of the patients with adrenal failure) were central type. Hypothyroidism profile showed (N = 9, 18%) that primary was more frequent than secondary cause (72% versus 22%). The secondary adrenal failure and hypothyroidism in addition to the mentioned cases with hypogonadotropic hypogonadism [22,29] and even one case with GH-IGF1 deficiency [32] represented the subjects with hypopituitarism (N = 11, 22%). (Figure 5).

Out of the 50 patients, nine (representing 18% from the entire cohort) had confirmed autoimmune endocrine diseases [18,20,23,26,30,33,34,36]. 22% of this group had only Addison’s disease [26,33] and 78% of them had at least two conditions (minimum one autoimmune endocrine ailment plus a second endocrine or non-endocrine autoimmune illness) [18,20,23,30,34,36]. Type 2 autoimmune polyendocrine syndrome was diagnosed in four cases, including the present one [18,23,36]); the second disease (other than thyroid or adrenal-related) was ulcerative colitis [20], autoimmune anemia [30], and (non-type2) diabetes [36] (Figure 6).

### 3.5. Proposed Workflow Algorithm in Endocrine Petrified Ear

A multidisciplinary team is required to assess a patient with petrified pinna, including radiologists, otorhinolaryngologists, dermatologists, surgeons, specialists in internal medicine and endocrinology. Which is the exact endocrine protocol of assessments under these circumstances is yet an open issue. Similarly, it is still an open matter whether dynamic testing should be done such as ACTH stimulation test to look for mild/partial forms of adrenal insufficiency of primary type, insulin tolerance test for seeking ACTH deficiency (secondary adrenal insufficiency) or repeating IGF1 assays in addition to performing an oral glucose tolerance test to check for acromegaly-related GH excess or testing twice the serum calcium and PTH level in order to perform the guideline definition of primary hyperparathyroidism diagnosis [134,135,136]. Also, what is the exact protocol of long- term follow-up in cases that were primarily diagnosed as idiopathic petrified ear is still an open question. Nevertheless, one patient might have several potential co-morbidities that are related to the auricular condition according to the current understanding of the ear disease. Probably, chronic renal failure should be re-classified into an endocrine form due to a high risk of associated endocrine and mineral metabolism anomalies [7,137,138,139]. We propose the term of endocrine petrified ear to highlight, on one hand, the increased risk that the patient might associate endocrine/metabolic ailments that potentially are connected to the pinna lesion and to stress one more time the importance of hormonal evaluation under these circumstances. 

Based on the mentioned data, we proposed a workflow endocrine algorithm to address the thyroid, adrenal, and pituitary function, on one hand, and the metabolic issues as well as mineral metabolism assays and renal function evaluation (Figure 7).

As limitations of the current analysis, we mention that this was a non-systematic review since the analyzed data were heterogeneous, and we did not intend to restraint the cases enrollment. Of note, the initial data were introduced according to the original reports, while the final endocrine cohort (N = 50) included the novel case and a re-interpretation of prior non-endocrine data according to our perspective of the topic. One of the main issues, as mentioned, remains the fact that the pathogenic rational behind petrified pinna still needs experimental studies to highlight the underlying mechanisms. Another aspect of approaching the auricular calcifications/ossifications is the fact that they may be underdiagnosed, and awareness helps the overall case management. 

## 4. Conclusions

To our aware this is one of the most complexes analyses of published modern data (N = 75) in the matter of endocrine petrified ear. Sample-based analyses showed that more than half of the patients have focal or extended ossificans which is a higher than previously reported. This might be caused by an increased level of awareness and a more detailed exploration of petrified pinna amid CT and histological report upon biopsy. Data coming from the two retrospective imagery-focused studies showed a rate varying between 7% and 23% with regard to petrified pinna, and this suggested a higher rate than expected according to prior data mostly coming from isolated cases reports. Approximately 18% of the patients underwent biopsy (mostly, incisional type) which remains a matter of individualized decision and, with three exceptions, surgery was not recommended. The longest duration post-diagnosis (of maximum of 7 years) did not show an improvement regardless of the treatment for the underlying condition. The novel case on point adds to the limited number of publications series with concern to the autoimmune poly-endocrine syndrome type 2 in petrified pinna. This is the longest duration of documented petrified ear (in terms of clinical, hormonal, thyroid autoimmunity, and imagery features) after first identification of the auricular calcification (of 10 years). Endocrine petrified ear stands on a complex profile of hormonal and autoimmunity issues that, according to our final analysis, revealed the most frequent types: adrenal insufficiency (36%, but less frequent Addison’s disease than central hypoadrenalism, opposite to prior data), hypopituitarism (22%), and hypothyroidism (18%, primary causes being found in 72% of them) while at least one autoimmune endocrine illness was confirmed in 18% of the individuals with petrified ear. The entity may be under-diagnosed so far and awareness is essential, particularly, from the endocrine perspective of associating a life- threatening condition such as adrenal insufficiency. Nevertheless, a complex multidisciplinary approach and acknowledgment represents the key of petrified ears.

## Figures and Tables

**Figure 1 diagnostics-14-01303-f001:**
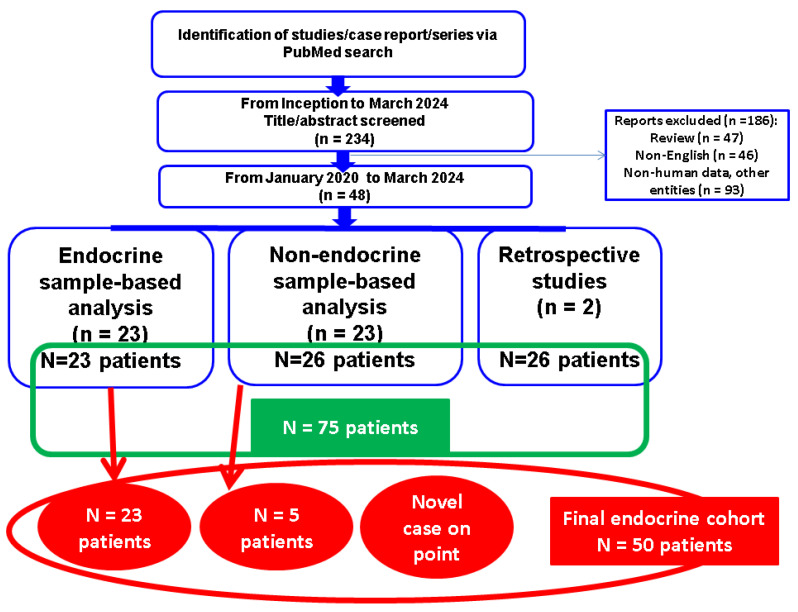
Workflow diagram according to our methods (n = number of studies or case reports/series; N = number of patients diagnosed with petrified ear).

**Figure 2 diagnostics-14-01303-f002:**
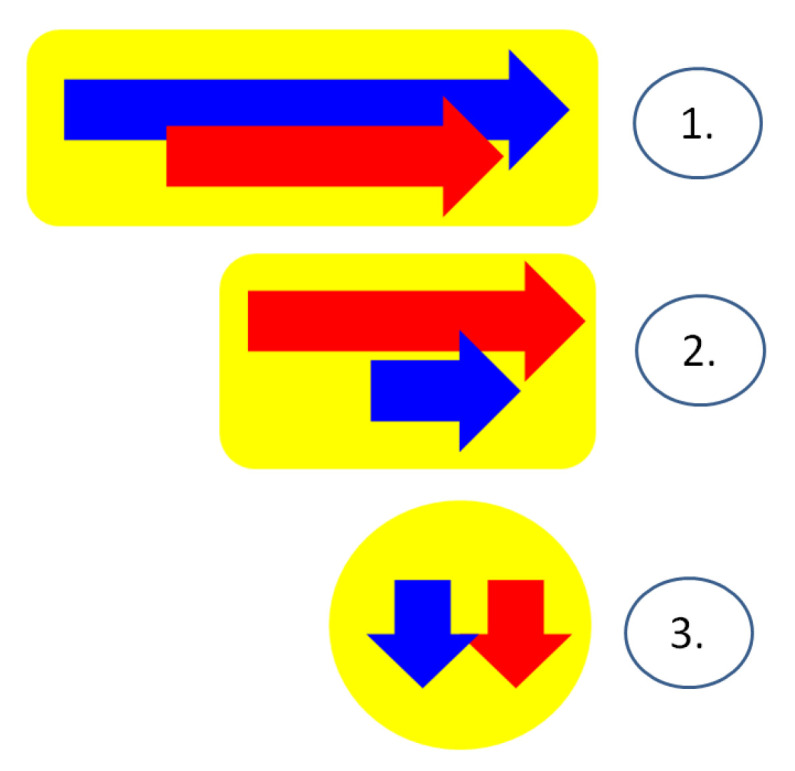
Patterns of interplay between the presence of an endocrine condition and the diagnosis of petrified ear: 1. The medical history includes a history of an endocrine disease and, later on, the self-palpation or palpation of rigid ears was done; 2. The medical history includes a longer period of time with progressive rigidity of the auricles followed by the diagnosis of endocrine elements; 3. The diagnosis of the endocrine anomaly was synchronously established with the confirmation of the petrified pinna [1,6,18,20,21,22,23,24,25,26,27,28,29,30,31,32,33,34,35,36,37,38,39] (red arrow = auricular calcification/ossification; blue arrow = endocrine disease).

**Figure 3 diagnostics-14-01303-f003:**
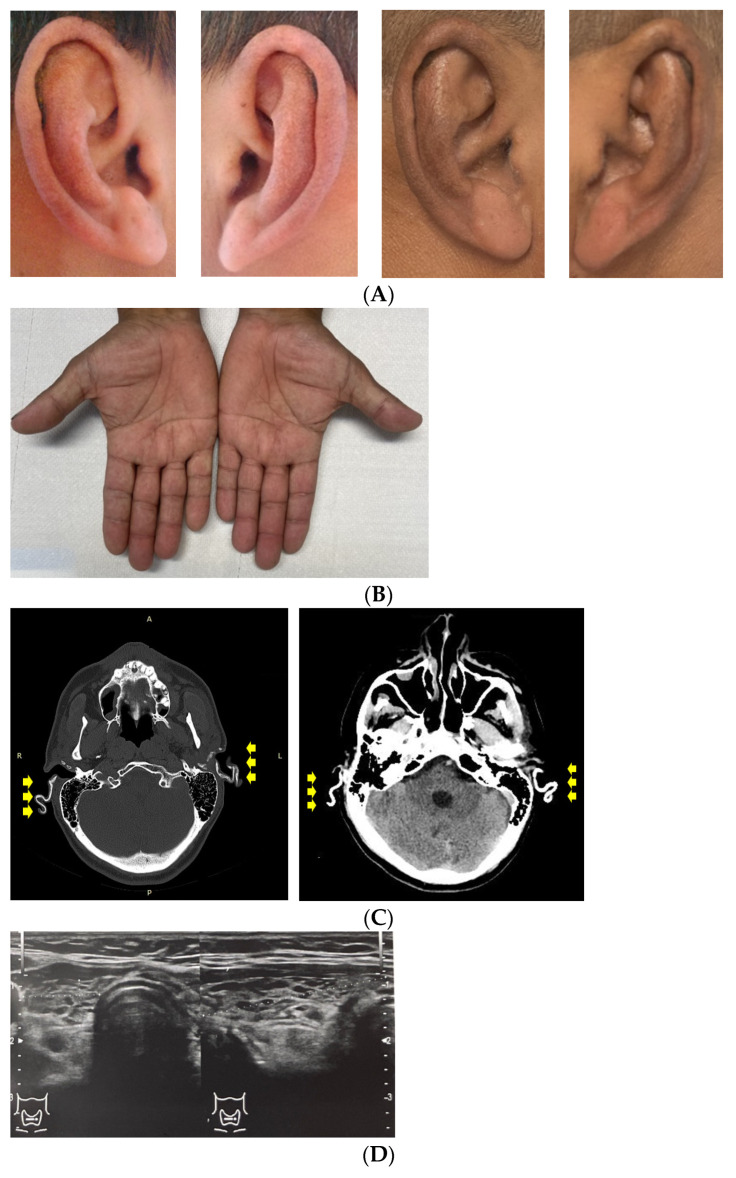
(**A**). Endocrine petrified ear: (left) a decade ago; (right) current evaluation; hyperpigmentation was associated with an adrenal crisis, and it was correlated with a poor therapy adherence for Addison’s disease; of note, the patient remained completely asymptomatic and the rigid pinna was accidentally identified during physical exam. (**B**). Hands hyperpigmentation: typical aspect for poorly controlled Addison’s disease. (**C**). CT scan (axial plan) showing bilateral calcified auricles (yellow arrow): (left) first diagnosis (a decade ago); (right) most recent evaluation. (**D**). Thyroid ultrasound at the most recent evaluation showing hypoechoic, inhomogeneous pattern suggestive for an autoimmune thyroid disease.

**Figure 4 diagnostics-14-01303-f004:**
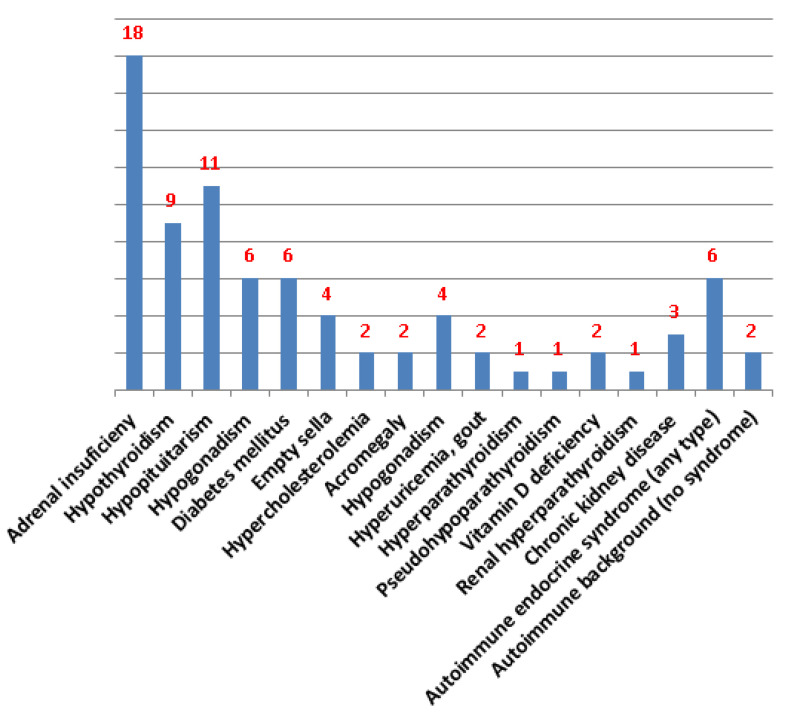
Number of patients with each of the endocrine/metabolic conditions (one patient might present more than one disease) according to our methods and the novel case [1,6,12,18,20,21,22,23,24,25,26,27,28,29,30,31,32,33,34,35,36,37,38,39,41,46,50,52].

**Figure 5 diagnostics-14-01303-f005:**
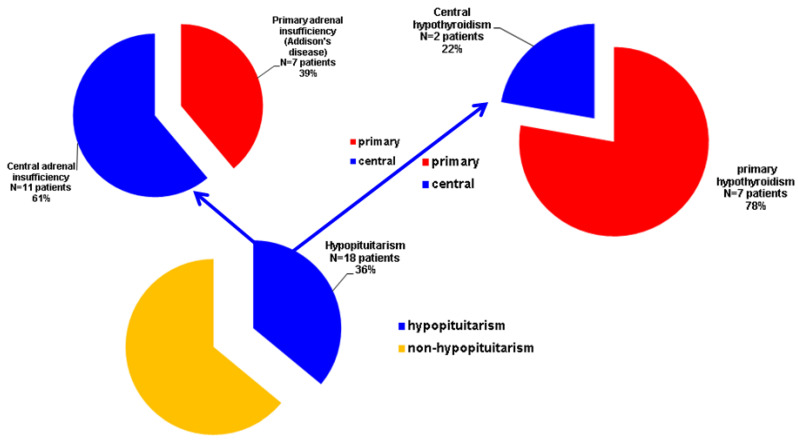
Analysis of types of adrenal insufficiency and hypothyroidism (primary versus secondary; secondary forms were included in the sub-group with hypopituitarism).

**Figure 6 diagnostics-14-01303-f006:**
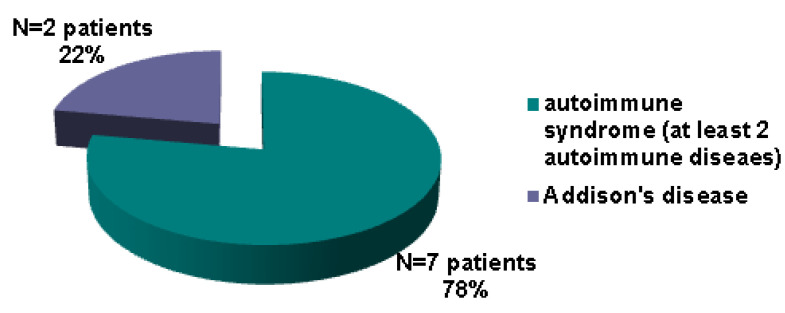
Quantitative analysis of autoimmune aspects in endocrine petrified ear (endocrine autoimmune elements were confirmed in nine out of 50 patients with pinna calcifications/ossifications).

**Figure 7 diagnostics-14-01303-f007:**
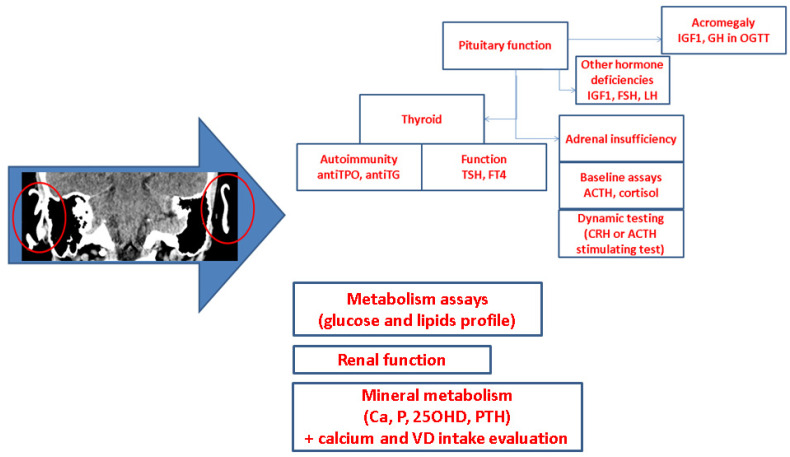
Proposed workflow algorithm to assess the potential endocrine and metabolic components in patients with petrified ear: thyroid function involves TSH (thyroid-stimulating hormone) and FT4 (free levothyroxine), and thyroid autoimmunity involves, namely, antiTPO (anti-thyroperoxidase antibodies) and antiTg (anti-thyroglobulin antibodies); the diagnosis of adrenal insufficiency requires plasma cortisol and ACTH (adrenocorticotropic hormone) assays and (if needed, not routinely) dynamic testing such as an ACTH stimulating test for primary adrenal failure and CRH (corticotropin-releasing hormone) testing for secondary type; pituitary function exploration includes the above mentioned assays and the exploration of other hormones like gonadotropin deficiency or GH (growth hormone) deficiency as reflected by a low blood level of IGF1 (Insulin-like Growth Factor 1); also, pituitary dysfunction should include GH excess (acromegaly) as revealed by high IGF1 and increased GH level amid OGTT (oral glucose tolerance test). In the meantime, each patient with rigid pinna should have an evaluation of glucose and lipid metabolism as well as renal function and biochemistry panel in terms of Ca (serum calcium) and P (serum phosphorus) associated with the mineral metabolism-related hormones PTH (parathormone) and 25OHD (25-hydroxyvitamin D) and calcium and VD (vitamin D) intake (if any) (the capture is based on the evaluation of the same patient presented in Section 3.3).

**Table 1 diagnostics-14-01303-t001:** Endocrine petrified ear: the published data with concern to the diagnosis of petrified ear that associated a prior or current diagnosis of an endocrine condition (autoimmune or not) according to our methods; the display starts with the most recent publication date [1,6,18,20,21,22,23,24,25,26,27,28,29,30,31,32,33,34,35,36,37,38,39].

First Author Reference Number	Year of Publication	Patient	B/U	Petrified Ear (PE) Features ▶ Clinical Detection/Presentation ▶ Imagery Scans ▶ Outcome	Endocrine Conditions: ▶ Endocrine Conditions ▶ Other Key Findings Co-Morbidities
Zhao [20]	2023	M, 57	▶ B	▶ Admission for ulcerative colitis diagnosis (diarrhea, abdominal pain, bloody stool) → detection of rigid pinna ▶ X-ray: ossification (+histological confirmation)	▶ Autoimmune thyroid disease (diagnosed during acute colitis episode) + ulcerative colitis
Awasthi [1]	2022	M, 43	▶ B	▶ Mild intermittent ear pain (years before the endocrine diagnosis) ▶ Confirmation via CT scan: >1100 HU at pinna (ossification) ▶ Whole body bone scintigraphy with 99 m-Tc MDP + SPECT: bilateral tracer uptake	▶ Acromegaly (treated with hypophysectomy, gamma knife + somatostatin analogue) Complicated with: ▶ Diabetes mellitus (therapy with metformin) ▶ Uncontrolled secondary adrenal failure ▶ Central hypothyroidism
Lee [21]	2022	M, 58	▶ B	▶ Admission for rigid pinna and bilateral stenotic external ear canals ▶ CT: calcification ▶ No hearing loss ▶ No intervention: follow-up for 7 years	▶ History of acromegaly and neurosurgery 24 years before ▶ Diabetes mellitus 14 years before
Recalcati [18]	2021	M, 71	▶ B	▶ Progressive rigid helix for 20 years + pain in bad at night + hearing loss ▶ Confirmation via X-ray, CT (partial ossification)	▶ 18-year history of primary hypothyroidism (treated with LT4) → normal function on admission ▶ + primary adrenal insufficiency (treated with hydrocortisone) ▶ Left ear: squamous cell carcinoma of scapha (surgical resection → reconstruction via two-stage interpolated, inferiorly based, pre-auricular transposition flap)
Kannenberg [22]	2021	M, 75	▶ B	▶ 20-year history of recurrent weakness ▶ 10-year history of rigid pinna ▶ X-ray, 3D-CT: ossification	▶ On admission: somnolence, hypotension, cachexia, hyponatremia, mild hyperkaliemia, low cortisol, low ACTH, negative adrenal antibodies → secondary adrenal insufficiency → glucocorticoids + mineralocorticoids replacement ▶ + central hypogonadism (no hypothyroidism) → good clinical evolution at 1-month control (stationary pinna aspect)
Sedhai [23]	2019	M, 47	▶ B	▶ Presentation for acute pericarditis with tamponade (hypotension, tachycardia, shock) ▶ Clinical exam detected rigid pinna ▶ Confirmation via brain CT scan (auricular calcification) + associated partial empty sella	▶ Type 2 polyglandular autoimmune syndrome (autoimmune hypothyroidism + Addison’s disease) complicated with polyserositis was diagnosed amid first admission for acute pericarditis ▶ Therapy: emergent pericardiocentesis → short course NSAI + colchicine + start LT4 and hydrocortisone → clinically stable after 6 months
Caton [24]	2019	M, 47	▶ B	▶ CT: calcification	▶ History of traumatic brain injury (3 years before) → secondary adrenal insufficiency
Taguchi [25]	2017	M, 22	▶ B	▶ Admission for acute adrenal insufficiency (nausea) ▶ Confirmation via X-ray and 3D-CT ▶ No specific therapy for PE	▶ Hyponatremia and low plasma cortisol → diagnosis of isolated deficiency of ACTH ▶ Good clinical outcome for 6 years (no remission of the PE)
James [26]	2016	M, 63	▶ B	▶ Admission for progressive hardening of both pinna ▶ X-ray confirmation (calcification + probable ossification) ▶ No specific therapy for PE	▶ Addison’s disease since the age of 12 years (controlled under hydrocortisone and fludrocortisone)
Goswami [27]	2016	M, 30	▶ B	▶ Clinical deterioration amid LT4 supplementation → suspicion of adrenal insufficiency (confirmed) ▶ On first admission: 1-year history of weakness, easy fatigability and loss of appetite ▶ X-ray, non-contrast CT confirmation of PE (calcification) ▶ Contrast-enhanced pituitary MRI: partial empty sella	▶ Primary hypothyroidism (started LT4) → detection of secondary adrenal insufficiency (probably a late sequel of prior acute meningitis) → good outcome under prednisolone, but similar aspect of PE
Calderón-Komáromy [28]	2015	M, 45	▶ B	▶ Admitted for hyponatremia ▶ 2-year history of rigid pinna ▶ X-ray: ossification ▶ Biopsy: foci of ossification (mature trabecular bone)	▶ Undetectable plasma cortisol on first admission ▶ Normal response to corticotropin test → central adrenal insufficiency of unknown cause (no endocrine autoimmunity)
Koning [29]	2014	M, 68	▶ B	▶ Peri-operatory (thymoma resection) onset of adrenal insufficiency (weight loss, lack of strength, hypotension)	▶ Hypopituitarism (adrenal insufficiency and hypogonadism)
Buikema [6]	2012	M, 79	▶ U	▶ Incidental finding at physical exam: rigid ear ▶ Clinical exam: skin with hyperpigmentation of superior helix ▶ 1-year history of hearing loss (left ear) ▶ Non-contrast CT scan confirmation (ossification) ▶ The patient declined biopsy (no other intervention was performed)	▶ History of primary hyperparathyroidism/hypercalcemia (5 years before the diagnosis of PE) → treated with subtotal parathyroidectomy (3 years before the diagnosis of PE) → current normal PTH/calcium ▶ Current mild microcytic anemia
Gogate [30]	2012	M, 53	▶ U	▶ 6-month history of diarrhea, weakness, fatigability, cold intolerance, mental disturbances ▶ 1-year history of rigid pinna ▶ Pituitary MRI: empty sella ▶ CT: auricular calcification	▶ Vitamin D deficiency with normal PTH ▶ Pernicious anemia (positive anti-parietal cell antibody; biopsy: non-specific gastritis) ▶ Primary autoimmune thyroiditis with hypothyroidism (positive antiTPO) ▶ Secondary adrenal insufficiency (possibly due to lymphocytic hypophysitis)
Uthoff [31]	2011	M, 60	▶ B	▶ Admission as emergency for hypotension → detection of rigid pinna ▶ X-ray, CT: calcification	▶ Secondary adrenal insufficiency (autoimmune hypophysitis)
Machado [32]	2009	F, 57	▶ B	▶ Admission for 2-day history of fever, headache, behavior changes (treated as meningitis + urinary infection) ▶ CT: confirmation of PE amid investigations for suspected meningitis ▶ Biopsy confirmation (ossification: lamellar bone, Haversian channels + bone marrow tissue) ▶ Pituitary MRI: small gland	▶ Post-partum hypopituitarism due to lymphocytic hypophysitis; this diagnosis was only established during current hospitalization despite prior pregnancy 15 years ago (first published case with PE) ▶ Low ACTH, cortisol, GH-RH, TSH (normal LH, FSH) ▶ Thyroid autoimmune disorder
Richter [33]	2009	M, 69	▶ B	▶ Rigid pinna detection during admission for a pulmonary infection amid prior diagnosis of (and therapy for) Addison’s disease ▶ X-ray: ossification	▶ Primary adrenal insufficiency
Mastronikolis [34]	2009	M, 75	▶ B	▶ Admission: mild pinna discomfort during sleeping + rigid pinna ▶ X-ray: ossification ▶ Excisional biopsy: ossification	▶ History of Addison’s disease and diabetes mellitus
Strauss [35]	2008	M, 62	▶ B	▶ Bilateral asymptomatic hardening of the pinna ▶ X-ray: calcification	▶ Pseudohypoparathyroidism (short fifth metacarpal bones bilaterally; slightly elevated PTH; short stature)
Thomson [36]	2007	M, 70 (C1)	▶ B	▶ Gradual stiffening of the ear and partial loss of left pinna starting with the age of 18 (frostbite) ▶ X-ray, CT: calcification	▶ Two family members had PE ▶ C1: PE was diagnosed years before the endocrine conditions (12-year history of hypothyroidism + 6-year history of Addison’s disease)
Chiu [37]	2006	F, 52	▶ B	▶ 6-hystory of progressive rigid pinna ▶ Pinna calcification	▶ Chronic kidney disease (hemodialysis) (a history of 13 years) ▶ Renal hyperparathyroidism: hypercalcemia, hyperphosphatemia, high PTH (3254 pg/mL) ▶ Rigid pinna remained the same 1 year after parathyroidectomy
High [38]	2004	M, 60	▶ B	▶ 10-year history of progressive rigid pinna ▶ X-ray: ossification ▶ Biopsy: ossification (trabecular bone)	▶ History of diabetes mellitus controlled under diet
Wang [39]	2002	M, 43	▶ B	▶ Bilateral PE (ossification)	▶ Hypopituitarism (secondary adrenal insufficiency)

Abbreviations: ACTH = adrenocorticotropic hormone; antiTPO = anti-thyroperoxidase antibodies; B/U = bilateral or unilateral ear involvement; 3D = three-dimensional; C1 = case 1 (patient from a case series); CT = computed tomography; F = female; FSH = follicle-stimulating hormone; GH-RH = growth hormone-releasing hormone; LH = luteinizing hormone; HU = Hounsfield units; MRI = magnetic resonance imagery; M = male; NSAI = non-steroidal anti-inflammatory drug; PE = petrified ear; PTH = parathormone; SPECT = single-photon emission CT; T4 = levothyroxine; Tc = technetium; TSH = thyroid-stimulating hormone.

**Table 2 diagnostics-14-01303-t002:** Sample-focused analysis on petrified ear in subjects apparently without any endocrine conditions (according to the original reports); the display starts with the most recent publication date [2,4,7,9,12,36,40,41,42,43,44,45,46,47,48,49,50,51,52,53,54,55,56,57].

First Author Reference Number	Year of Publication	Patient	B/U	Petrified Ear (PE) Features ▶ Clinical Detection/Presentation ▶ Imagery Data ▶ Biopsy + Pinna Intervention (If Any)	Cause of Petrified Ear + Other Co-Morbidities
Reddy [2]	2023	M, 63	▶ B	▶ 2-year history of pinna stiffening + local pain during sleeping ▶ Bilateral mild high-frequency hearing loss (age-related) ▶ High Resolution CT confirmation	▶ Idiopathic PE
Arora [40]	2020	M, 23	▶ B	▶ One patient out of a 5-patient series diagnosed with Primrose syndrome ▶ Hearing loss + ear cartilage calcification	▶ Primrose syndrome (specific facies, learning disabilities, platybasia, bitemporal narrowing, mild platyspondyly, genu valgum, thoracic kyphosis)
Thomas [41]	2019	M, 52	▶ B	▶ Bilateral PE, unilateral fracture in PE ▶ Fracture in right PE (first fracture report in PE)—manifested with 1-year history of right helix pain (increased pain during washing and sleeping) ▶ Clinical exam and X-ray confirmation: bilateral PE (+ossification) ▶ Surgery: declined by the patient ▶ Spontaneous right ear pain remission after 1 year	▶ 20 years before: bilateral ear frostbite ▶ Hypercholesterolemia (treated with atorvastatin) ▶ Mild learning disability (therapy with amitriptyline)
Shah [4]	2019	M, 75	▶ U	▶ 6-month history of hardening and protrusion of the right ear ▶ Mild local erythema ▶ X-ray confirmation ▶ Incisional biopsy: ossification	▶ Idiopathic PE ▶ History of basal cell carcinoma + seborrheic dermatitis
Harker [42]	2019	M, 60	▶ B	▶ 10 to 20-year history of rigid pinna ▶ X-ray confirmation: calcification ▶ Incisional biopsy: auricular ossificans	▶ Idiopathic PE
Weiss [12]	2017	M, 69	▶ B	▶ History of melanoma in situ → he was admitted for routine check-up ▶ 1-month history of non-painful hardening of right ear → then left year ▶ X-ray confirmation (ossification) ▶ No intervention for PE (recommendation of orthotic pillow)	▶ Idiopathic PE ▶ History of melanoma in situ ▶ Chronic kidney disease stage III (caused by hypertension) ▶ Hypovitaminosis D
Mohan [43]	2017	M, 59	▶ B	▶ Admission for chronic obstructive pulmonary disease ▶ Episodic pain and redness of the ear lobes ▶ X-ray confirmation (calcification)	▶ Relapsing polychondritis ▶ Neutrophilic leucocytosis ▶ Chronic normocytic anemia ▶ Negative for antinuclear and antineutrophil cytoplasmic antibodies
Gottesman [44]	2016	M, 69 (P1) M, 20 (P2) F, 18 (P3)	▶ B	▶ Three out of four patients with juvenile Paget’s disease and PE (ossification) ▶ P1: admission at the age of 60 after 4 years of bisphosphonates; PE since the age of 45, diagnosis of PE at the age of 60 ▶ P2: long term bisphosphonates therapy ▶ P3: hearing loss	▶ Osteoprotegerin-deficiency juvenile Paget disease type 1
Thorne [45]	2016	F, 29	▶ B	▶ 9-year history of relapsing polychondritis ▶ Self-palpation of rigid pinna ▶ X-ray confirmation: calcification	▶ Relapsing polychondritis
Karrs [46]	2016	M, his 70s	▶ B	▶ Incidental finding ▶ CT confirmation ▶ Incisional biopsy: calcification + ossification	▶ Idiopathic PE Co-morbidities: ▶ Type 2 diabetes mellitus ▶ Chronic kidney disease ▶ Hypertension ▶ Gout
Aw [7]	2015	F, 73	▶ B	▶ Admission for conductive hearing loss ▶ Otoscopic exam: otoscope did not pass into left external auditory canal ▶ High Resolution temporal CT: calcifications	▶ Idiopathic PE
Khosroshahi [47]	2014	F, 37	▶ B	▶ Bilateral PE (ossification)	▶ Keutel syndrome
Anchan [48]	2013	F, 20	▶ B	▶ Admission for exertional breathlessness, nasal obstruction, and rigid pinna ▶ X-ray, CT: calcification of pinna and tracheobronchial tree	▶ Relapsing polychondritis
Alsey [49]	2012	M, 40	▶ B	▶ Unilateral hearing loss ▶ Incidental finding	▶ Idiopathic PE
Chang [50]	2011	F, 72	▶ U	▶ Progressive left helix rigidity (within months) + local pain during sleeping ▶ CT confirmation: ossification ▶ Incisional biopsy	▶ Idiopathic PE ▶ Current mild anemia ▶ 8-year history of daily calcium + vitamin D intake for osteoporosis (current normal phosphorus + calcium levels) ▶ 10-year history of diabetes, high blood pressure, angina pectoralis
Kim [51]	2011	M, 53	▶ U	▶ X-ray: ossification	▶ Post-local trauma (rubbing the pinna)
Posmyk [52]	2011	M, 27	▶ B	▶ Bilateral PE (ossification) ▶ Suggestive phenotype: muscle wasting, brain calcification, mild intellectual disability, hearing loss, cataract	▶ Primrose syndrome (the sixth and the youngest published case of Primrose syndrome in 2011) ▶ Hypergonadotropic hypogonadism → osteoporosis
Britton 9 [9]	2009	M, 28	▶ U	▶ Admission for right ear with local swelling, tenderness, and firmness after he used Bluetooth headphones for 5 months (at least 6 h per day, 5 days per week) ▶ Local pressure was exacerbated amid cold exposure and using a stocking cap	▶ Idiopathic PE
Laguna [53]	2009	M, 65	▶ B	▶ Bilateral (asymmetrical) PE ▶ Progressive right helix rigidity (10-year history) ▶ Confirmation via X-ray ▶ Biopsy (calcification + foci of ossification)	▶ Idiopathic PE
Carfrae [54]	2008	M, 49	▶ U	▶ Admission for left hearing loss + stenotic left external auditory meatus ▶ CT: ossification (cartilaginous auricle +external auditory canal) ▶ Treatment: meatoplasty (pathological report: osseous metaplasia)	▶ Idiopathic PE
Sterneberg-Vos [55]	2007	M, 72	▶ U	▶ Admission for 7-month history of swollen, rigid, painful during sleep right pinna ▶ X-ray: ossification ▶ Biopsy was attempted, but was not successful due to the ear rigidity ▶ Wedge excision of the rigid upper rim of the right auricle with clinical improvement	▶ Prior exposure to hypothermia (22 years before)
Thomson [36]	2007	F, 30 (C2)	▶ B		▶ Two family members (C1′s daughter) ▶ C2: no endocrine condition
Parmar [56]	2007	F, 9	▶ B	▶ Admission for hearing loss since the age of 15 months ▶ CT confirmation: calcification	▶ Keutel syndrome
Manni [57]	2005	F, 63	▶ B	▶ Admission for hearing loss (ossification of pinna + external ear canal) ▶ Surgical resection of the cartilage external ear canal and tragus → post-operatory histological report: ossification (Haversian canals + bone marrow)	▶ Idiopathic PE

Abbreviations: B/U = bilateral or unilateral PE; C1, C2 = case 1 or 2 (patients from case series); CT = computed tomography; F = female; M = male; PE = petrified ear; P1, P2, or P3 = patient 1, 2, or 3 (patients from case series).

**Table 3 diagnostics-14-01303-t003:** Studies to address the identification of auricular calcifications/ossifications amid CT scans according to our methods [7,15].

First author Reference Number	Year of Publication	Study Design	Studied Population	Main Findings
Aw [7]	2015	retrospective study	▶ 51 consecutive patients who underwent temporal bone CT scan (in one year—2007)	▶ N = 12 patients had foci or extensive auricular calcifications
Gossner [15]	2014	retrospective study	▶ 200 patients who underwent consecutive head CT scans ▶ N = 19.5% (N = 39) had auricular calcifications/ossifications of the cartilage from external ear (all of them were incidentally detected and asymptomatic) (N1 + N2 + N3) ▶ N1 + N2 represents PE (N = 14)	▶ N1 = 13 (6.5%) patients with affected cartilage of the auricle ▶ N3 = 25 (12.5%) patients with affected external auditory channel ▶ N1 = one patient had both affected sites (0.5%)

Abbreviations: CT = computed tomography; N = number of patients; N1, N2, N3 = patients sub-groups; PE = petrified ear.

**Table 4 diagnostics-14-01303-t004:** Hormonal panel on first detection of the petrified pinna and 10 years later.

Parameter	At the Age of 42 *	At the Age of 52 **	Normal Values
TSH (µUI/mL)	7.55	7.8	0.4–4
FT4 (ng/dL)	1.1	0.79	0.89–1.76
antiTPO (UI/mL)	66	14 ***	<50
antiTg (UI/mL)	12	11	<60
ACTH (pg/mL)	678	1031	7.2–63.3
Total cholesterol (mg/dL)	288	296	<200
Uric acid (mg/dL)	6	9.5	3.5–7.2

Abbreviations: ACTH = adrenocorticotropic hormone; antiTPO = anti-thyroperoxidase antibodies; antiTg = anti-thyroglobulin antibodies; FT4 = free levothyroxine; TSH = thyroid stimulating hormone; * on admission, the patient was treated daily 10 mg prednisone and 0.1 mg fludrocortisone and, after intravenous hydrocortisone, an increase in prednisone dose with 5 mg/day was offered to the subject in addition to starting levothyroxine 25 µg/day; ** the patient was under same regime for chronic adrenal insufficiency and 100 µg of daily levothyroxine; *** normalization of the serum antibodies against thyroid did not exclude a Hashimoto thyroiditis with suggestive ultrasound features.

**Table 5 diagnostics-14-01303-t005:** Endocrine perspective in petrified pinna: endocrine petrified ear according to our methods and the novel case on point [1,6,12,18,20,21,22,23,24,25,26,27,28,29,30,31,32,33,34,35,36,37,38,39,41,46,50,52].

First Author Reference Number	Patient	HypoT	Central or Primary	AI	Secondary or Primary AI	Other Endocrine-Metabolic Diseases	Endocrine Autoimmunity (If Specified)	Non-Endocrine Autoimmune Disease
Zhao, 2023 [20]	M, 57	hypoT	primary				YES (APS)	Ulcerative colitis
Awasthi, 2022 [1]	M, 43	hypoT	central	AI	secondary	acromegaly hypopituitarism DM		
Lee, 2022 [21]	M, 58					acromegaly DM		
Recalcati, 2021 [18]	M, 71	hypoT	primary	AI	primary		YES (APS2)	
Kannenberg, 2021 [22]	M, 75			AI	secondary	hypopituitarism (ACTH deficiency and central hypogonadism)		
Sedhai, 2019 [23]	M, 47	hypoT	primary	AI	primary	partial empty sella	YES (APS2)	
Caton, 2019 [24]	M, 47			AI	secondary	hypopituitarism (post-traumatic)		
Taguchi, 2019 [25]	M, 22			AI	secondary	hypopituitarism (isolated ACTH deficiency)		
James, 2016 [26]	M, 63			AI	primary		YES	
Goswami, 2016 [27]	M, 30	hypoT	primary	AI	secondary	hypopituitarism (isolated ACTH deficiency) empty sella		
Calderón-Komáromy, 2015 [28]	M, 45			AI	secondary	hypopituitarism (isolated ACTH deficiency)		
Koning, 2014 [29]	M, 68			AI	secondary	hypopituitarism (ACTH deficiency and central hypogonadism)		
Buikema, 2012 [6]	M, 79					hyperPT		
Gogate, 2012 [30]	M, 53	hypoT	primary	AI	secondary	hypopituitarism (isolated ACTH deficiency) empty sella VD deficiency (normal PTH)	YES (APS)	Pernicious anemia
Uthoff, 2011 [31]	M, 60			AI	secondary	hypopituitarism (isolated ACTH deficiency)	YES	
Machado, 2009 [32]	F, 57	hypoT	secondary	AI	secondary	hypopituitarism (prior postpartum lymphocytic hypophysitis) + low GH-IGF1, no hypogonadism	YES	
Richter, 2009 [33]	M, 69			AI	primary		YES	
Mastronikolis, 2009 [34]	M, 75			AI	primary	DM	YES	
Strauss, 2008 [35]	M, 62					Pseudohypoparathyroidism (slightly increased PTH)		
Thomson, 2007 [36]	M, 70	hypoT	primary	AI	primary		YES (APS2)	
Chiu, 2006 [37]	F, 52					secondary (renal hyperPTH) CKD (high calcium, phosphorus, PTH)		
High, 2004 [38]	M, 60					DM	NO	
Wang, 2002 [39]	M, 43			AI	secondary	hypopituitarism (ACTH deficiency)	NO	
**Re-classification for the (initial) non-endocrine cohort**								
Thomas, 2019 [41]	M, 52					hypercholesterolemia		
Weiss, 2017 [12]	M, 69					CKD (caused by hypertension) VD deficiency		
Karrs, 2016 [46]	M, his 70s					type 2 DM CKD (hypertension) gout		
Chang, 2011 [50]	F, 72					calcium + VD therapy for osteoporosis DM hypertension		
Posmyk, 2011 [52]	M, 27					hypergonadotropic hypogonadism → osteoporosis		
**Novel case on point**								
Present case, 2024	M, 52	hypoT	primary	AI	primary	empty sella hypercholesterolemia hyperuricemia	YES (APS2)	

Abbreviations: AI = adrenal insufficiency; APS = autoimmune polyendocrine syndrome; APS2 = autoimmune polyendocrine syndrome; ACTH = adrenocorticotropic hormone; CKD = chronic kidney disease; DM = diabetes mellitus; GH = growth hormone; hypoT = hypothyroidism; hyperPTH = hyperparathyroidism; IGF1 = insulin-like growth factor; PTH = parathormone; VD = vitamin D.

## Data Availability

Not applicable.

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
