# Peer review of "Endocrine Petrified Ear: Associated Endocrine Conditions in Auricular Calcification/Ossification (A Sample-Focused Analysis)"

_diagnostics, 2024, doi:10.3390/diagnostics14121303_

Round 1
Reviewer 1 Report
Comments and Suggestions for Authors
The Authors report a novel case and review of the recent literature.
Lines 104-105: “Some authors consider a certain age-related pattern...”. Do “some authors” corresponde to references 2,19?
Line 128: parenthesis is missing
Lines 122-114: "Until now, no distinct indications are listed to surgically approach the condition, and thus the topic remains at individual decision in relationship with ear changes and the underlying general condition (if any)." I would speak more generally, not just talking about surgery, but talking about treatment in general.
Table 1. and Table 2. These tables are not so esay and quick to read. I would create 2 separate columns in each table, 1 with “bilateral or unilateral”, 2 wih “diagnostic findings” (e.g. x-ray or CT or histology..)
Lines 248-253: is it possibile to differentiate calcifications from ossifications by x-rays? Some Authors say that calcifications appear as deposits of increased density, while ossifications may show signs of actual bone tissue formation, such as trabeculae or cortices.
Lines 299: Could be helpful discussing about limitations, it's important to consider factors like speed, cost, resource availability, and technical complexity of imaging techniques.
Line 446: “The calcification amid petrified ears should be differentiated by isolated nodules in calcinosis cutis which is found more frequently in pediatric population” change with "Calcification in petrified ears should be distinguished from isolated nodules found in calcinosis cutis, which is more common in pediatric populations."
Line 469-471: “Secondarily, in people with petrified pinna and endocrine conditions, we might expect that, in case of a clear pathogenic connection, the control of the hormonal anomaly improves the auricular lesion, an aspect which was not confirmed so far.” Do we really expect the petrified ear to be a reversible condition?
Lines 752-754: "The burden of an endocrine disease or of a combination of hormonal anomalies might complicate with calcified pinna, yet, with incompletely understood significance and less clear management." change with “Endocrine diseases or hormonal imbalances may lead to calcified pinna, but their significance and management remain incompletely understood”.
Lines 761-762: "Type 1 (monogenic, autosomal recessive) has an early; the subjects harbor pathogenic variants of the AIRE gene." change with "Type 1 (monogenic, autosomal recessive) typically manifests early, with affected individuals carrying pathogenic variants of the AIRE gene."
Comments on the Quality of English Languagelook at the previous box
Author Response
Response to Review 1 Comments
Dear Reviewer,
Thank you very much for your time and your effort to review our manuscript.
We are very grateful for providing your valuable feedback on the article.
Here is our response and related amendment that has been made in the manuscript according to your review (marked in yellow color).
The Authors report a novel case and review of the recent literature.
Lines 104-105: “Some authors consider a certain age-related pattern...”. Do “some authors” corresponded to references 2,19?
Thank you very much. We corrected it. The entire paragraph corresponds to the references 2 to 19. Thank you
Line 128: parenthesis is missing
Thank you very much. We corrected it. The parenthesis was unnecessary and thus it was removed. Thank you
Lines 122-114: "Until now, no distinct indications are listed to surgically approach the condition, and thus the topic remains at individual decision in relationship with ear changes and the underlying general condition (if any)." I would speak more generally, not just talking about surgery, but talking about treatment in general.
Thank you very much. We followed your recommendation and added “general treatment”. Thank you
Table 1. and Table 2. These tables are not so essay and quick to read. I would create 2 separate columns in each table, 1 with “bilateral or unilateral”, 2 with “diagnostic findings” (e.g. x-ray or CT or histology..)
Thank you very much. We followed your recommendation and introduced a new column with “B/U” (the fourth column in Table 1 and 2). With regard to the “diagnostic findings”, these data are not homogenous amid the case reports (the petrified ear may be clinically detected as first step or during imagery evaluation or some imagery data have not been provided in other cases). These tables already represents a schematic and at glance approach of a large heterogeneous panel of parameters We tried the approach with an additional column, but in this way the column number 5 remains empty in some cases, hence, we respectfully mention that we incorporated column number 5 with the mentioned findings. Thank you very much
Lines 248-253: is it possibile to differentiate calcifications from ossifications by x-rays? Some Authors say that calcifications appear as deposits of increased density, while ossifications may show signs of actual bone tissue formation, such as trabeculae or cortices.
Thank you very much. This is not unanimously agreed. Moreover, once clinically or radiologically suspected, a CT scan was done in most cases amid modern era. Indeed, at CT scan, a distinction between these traits is feasible for experts as we mentioned. Thank you very much.
Lines 299: Could be helpful discussing about limitations, it's important to consider factors like speed, cost, resource availability, and technical complexity of imaging techniques.
Thank you very much. We followed your recommendation and added: “As limitations of the imagery evaluation we mention several factors such as the speed of diagnosis, associated cost of performing new assessments, resource availability, and technical complexity of the imaging techniques depending on the health care center.” Thank you
Line 446: “The calcification amid petrified ears should be differentiated by isolated nodules in calcinosis cutis which is found more frequently in pediatric population” change with "Calcification in petrified ears should be distinguished from isolated nodules found in calcinosis cutis, which is more common in pediatric populations."
Thank you very much. We corrected it. Thank you
Line 469-471: “Secondarily, in people with petrified pinna and endocrine conditions, we might expect that, in case of a clear pathogenic connection, the control of the hormonal anomaly improves the auricular lesion, an aspect which was not confirmed so far.” Do we really expect the petrified ear to be a reversible condition?
Thank you very much. We corrected it as follows: “Secondarily, in people with petrified pinna and endocrine conditions, we might expect that, in case of a clear pathogenic connection, the control of the hormonal anomaly limits the auricular lesion progression, an aspect which was not confirmed so far.” Thank you
Lines 752-754: "The burden of an endocrine disease or of a combination of hormonal anomalies might complicate with calcified pinna, yet, with incompletely understood significance and less clear management." change with “Endocrine diseases or hormonal imbalances may lead to calcified pinna, but their significance and management remain incompletely understood”.
Thank you very much. We corrected it. Thank you
Lines 761-762: "Type 1 (monogenic, autosomal recessive) has an early; the subjects harbor pathogenic variants of the AIRE gene." change with "Type 1 (monogenic, autosomal recessive) typically manifests early, with affected individuals carrying pathogenic variants of the AIRE gene."
Thank you very much. We corrected it. Thank you
Comments on the Quality of English Language: look at the previous box
Thank you very much. We followed your interesting and useful recommendations.
Thank you very much.
Reviewer 2 Report
Comments and Suggestions for Authors
Dear Editor in Chief,
Thanks for giving me the opportunity to review this interesting review ID diagnostics-3015110 entitled “Endocrine petrified ear: the myriad of associated endocrine conditions
in auricular calcification/ossification (a sample-focused analysis”. The article is interesting but has some design and methodological pitfalls. Attached below my comments that I hope would add to the manuscript.
Comments to the Authors,
I read with great interest the manuscript review ID diagnostics-3015110 entitled “Endocrine petrified ear: the myriad of associated endocrine conditions in auricular calcification/ossification (a sample-focused analysis”. It’s an interesting review with many precious data. My major concern with the paper is that the research methodology and quality of evidence is not mentioned in details.
Major comments:
Methodology:
It would be nice to add the methodology and if scoping review was used for analyzing the reliability of evidence for Endocrine petrified ear.
It would be nice also to add how the research question was formulated, it’s PIPOH and what was the search strategy.
It would be nice to add the Mesh words used and the serach engines used.
Would the authors clarify how data collection was done and if study design appraisal was done to the chosen papers.
Results:
It would be nice to add the evidence of the study outcomes.

Minor English editing is required.
Author Response
Response to Review 2 Comments
Dear Reviewer,
Thank you very much for your time and your effort to review our manuscript.
We are very grateful for your insightful comments and observations, also, for providing your valuable feedback on the article.
Here is a point-by-point response and related amendments that have been made in the manuscript according to your review (marked in yellow color).
Comments to the Authors,
I read with great interest the manuscript review ID diagnostics-3015110 entitled “Endocrine petrified ear: the myriad of associated endocrine conditions in auricular calcification/ossification (a sample-focused analysis”. It’s an interesting review with many precious data. My major concern with the paper is that the research methodology and quality of evidence is not mentioned in details.
Thank you very much. We provided the explanations below.
Major comments:
Methodology:
It would be nice to add the methodology and if scoping review was used for analyzing the reliability of evidence for Endocrine petrified ear.
It would be nice also to add how the research question was formulated, it’s PIPOH and what was the search strategy.
It would be nice to add the Mesh words used and the serach engines used.
Would the authors clarify how data collection was done and if study design appraisal was done to the chosen papers.
Results:
It would be nice to add the evidence of the study outcomes.
Thank you very much. This was a comprehensive (narrative) type of review.
The database of research: PubMed
The time frame of research: January 2000 to March 2024
The keywords of research: “calcified ear” (alternatively, “petrified ear”, “calcified pinnae”, “stony ear”, and “calcification of auricular cartilages”
The inclusion criteria: freely available, full-length articles varying from case reports to original studies
The exclusion criteria: reviews, non-human data, and non-English papers
The level of statistical evidence included in Results sections: Table 1 and 2 incldued case reports and series (the patients from case series are highlighted as “C1”, “C2” or “P1”, “P2”. Table 3 is specifically designated for studies.
PIPOH analysis was not feasible due to the heterogeneity of data collection that is why we choose a more flexible approach in order to gather more information. Hence, to our aware, this is one of the largest analyses published in the matter of endocrine petrified ear. Due to the heterogeneity of the spectrum in this specific matter, we choose to introduce the data as a non-systematic review since various levels of statistical evidence and various panels of parameters/data are identified in the mentioned papers. On the other hand, a systematic review pinpoints a specific critical assessment which in this matter is rather limited so far according to current scientific literature. However, this type of review is a well-recognized, standard, traditional approach which is suitable for topics with less generous publications such as the endocrine petrified ear. This allowed us to examine and evaluate the scientific panel on the specific area in a useful way for various practitioners, not only an endocrine perspective, across a practical multidisciplinary approach. Thank you very much.
Comments on the Quality of English Language: Minor English editing is required.
Thank you. We corrected it. Thank you
Thank you very much.